# Plasma Metabolomics Reveals Systemic Metabolic Remodeling in Early-Lactation Dairy Cows Fed a *Fusarium*-Contaminated Diet and Supplemented with a Mycotoxin-Deactivating Product

**DOI:** 10.3390/toxins18010009

**Published:** 2025-12-22

**Authors:** Gabriele Rocchetti, Alessandro Catellani, Marco Lapris, Nicole Reisinger, Johannes Faas, Ignacio Artavia, Silvia Labudova, Erminio Trevisi, Antonio Gallo

**Affiliations:** 1Department of Animal Science, Food and Nutrition, Università Cattolica del Sacro Cuore, Via Emilia Parmense 84, 29122 Piacenza, Italy; 2dsm-firmenich, Animal Nutrition and Health R&D Center, 3430 Tulln, Austria; 3Romeo and Enrica Invernizzi Research Center for Sustainable Dairy Production of the Università Cattolica del Sacro Cuore (CREI), 29122 Piacenza, Italy

**Keywords:** mycotoxin deactivation, plasma metabolomics, sphingolipid metabolism, ZENzyme^®^

## Abstract

This study investigated the systemic metabolic effects of feeding a *Fusarium*-contaminated diet to early-lactation Holstein cows, with or without a mycotoxin-deactivating product (MDP; Mycofix^®^ Plus, BIOMIN Holding GmbH, Tulln, Austria). Thirty cows were divided into three dietary groups: a mildly contaminated control (CTR), a moderately contaminated diet containing zearalenone and deoxynivalenol (MTX), and the same contaminated diet supplemented with MDP. Plasma collected at 56 days in milk was analyzed by untargeted ultra-high-performance liquid chromatography (UHPLC) coupled with high-resolution mass spectrometry (HRMS), and multivariate models identified discriminant metabolites and pathways. MTX-fed cows showed alterations in sphingolipid metabolism, including accumulation of ceramide (t18:0/16:0), lactosylceramide, and sphinganine 1-phosphate, consistent with ceramide synthase inhibition and lipid remodeling stress. Increases in estradiol, estrone, and cholesterol sulfate suggested endocrine disruption, while elevated 8-oxo-dGMP indicated oxidative DNA damage. MDP supplementation mitigated these alterations, reducing sphingolipid intermediates, modulating tryptophan and glycerophospholipid pathways, and lowering oxidative stress markers. Metabolites such as riboflavin, pipecolic acid, and N-acetylserotonin could be likely associated with an improved mitochondrial function and redox homeostasis, although future studies are required to confirm this hypothesis. Additionally, MDP-fed cows exhibited distinct shifts in pyrimidine and nucleotide metabolism. Overall, MDP effectively counteracted *Fusarium*-related metabolic disturbances, supporting its protective role in maintaining lipid balance, hormonal stability, oxidative control, and metabolic resilience.

## 1. Introduction

The *Fusarium*-produced mycotoxins are frequent contaminants of cereal-based feed ingredients commonly used in dairy cattle diets, particularly in total mixed rations (TMR) [1,2]. These secondary fungal metabolites, including deoxynivalenol (DON), zearalenone (ZEN), and fumonisins (FUM), are known to impair gut integrity, immune responses, and overall performance in ruminants, even at levels below regulatory guidance thresholds [3,4]. Although the rumen has traditionally been considered an effective barrier against dietary mycotoxins, increasing evidence suggests that early-lactation dairy cows may be more susceptible to their subclinical effects due to altered rumen function, high metabolic demands, and negative energy balance [5,6]. During this stage, physiological factors such as increased gut permeability, accelerated digesta passage, and reduced ruminal detoxification efficiency may allow a greater proportion of DON and ZEN to escape microbial biotransformation, thereby increasing the likelihood of systemic exposure. These conditions make early-lactation cows a particularly relevant model for assessing field-level *Fusarium* mycotoxin challenges and their subtle metabolic consequences.

To mitigate the adverse effects of mycotoxins, various detoxification strategies have been proposed, including the use of mycotoxin-deactivating products (MDPs) that rely on adsorptive, enzymatic, and microbial mechanisms [7]. These products are designed to reduce the bioavailability and toxicity of mycotoxins either by binding them in the gastrointestinal tract or by catalyzing their structural degradation into non-toxic metabolites [8]. In dairy production systems, MDPs are increasingly used as a preventive strategy, especially when feed contamination levels are within regulatory limits but still capable of eliciting subclinical effects that impair cow health, fertility, and productivity. Previous studies have reported improvements in feed intake, milk yield, and immune responses following MDP supplementation, yet the underlying immunometabolic adaptations remain largely unexplored [2,9]. Understanding how these products influence systemic metabolism is essential for optimizing their use and evaluating their efficacy beyond conventional performance parameters.

In this context, untargeted metabolomics offers a powerful tool to capture holistic changes in the plasma metabolome, providing novel insights into host responses to dietary toxins and feed additives [7]. In particular, the plasma metabolome can reveal early biochemical perturbations related to energy metabolism, amino acid (AA) turnover, oxidative stress, and lipid remodeling, all critical processes in the early lactation period [10]. Recently, a metabolomics approach has been used to evaluate the impact of mycotoxins, such as DON, on rumen function, production, and health of dairy cows [11]; the authors demonstrated that high DON concentrations in the diet of lactating cows significantly reduce both the microbial nitrogen metabolism and the protein content of milk. Another study [12] evaluated changes in milk metabolomic profiles resulting from a mycotoxin-contaminated corn silage intake by dairy cows; particularly, the authors provided new insights and novel milk biomarkers, such as sphingolipids, purines, pyrimidines, and oxidized glutathione, as a function of high and low levels of *Aspergillus*, *Penicillium*, and *Fusarium* mycotoxins. Interestingly, by using a metabolomic approach based on plasma ^1^H NMR, Wang et al. [13] identified eight significant metabolites as a function of mycotoxin contamination that were mainly involved in the metabolic pathways related to aminoacyl-tRNA biosynthesis, nitrogen metabolism, and alanine, aspartate, and glutamate metabolism.

Therefore, building on this background, the present study specifically aimed to compare the plasma metabolome of Holstein dairy cows in early lactation fed a mildly contaminated control diet versus a *Fusarium*-contaminated diet rich in ZEN and DON, and to evaluate whether supplementation with a commercial MDP could mitigate these effects. From the same trial, Catellani et al. [14] recently showed that a *Fusarium* mycotoxin contamination could modify the physiological post-calving resumption of ovarian activity, alter the physiological follicular development, and increase the occurrence of follicular cysts. Interestingly, the MDP was reported to reduce some negative mycotoxin effects on milk production and quality. Therefore, by integrating high-resolution mass spectrometry-based untargeted metabolomics and multivariate statistical analyses, we sought to characterize systemic metabolomic shifts associated with mycotoxin challenge and evaluate the potential protective role of dietary intervention.

## 2. Results

### 2.1. Plasma Metabolome and Multivariate Statistical Discrimination

The untargeted metabolomic approach enabled the putative annotation of 2015 mass features using the comprehensive Bovine Metabolome Database. Notably, 159 plasma metabolites were structurally confirmed via tandem MS/MS by leveraging information from the pooled quality control (QC) samples. Under our analytical conditions, 504 metabolites showed relative standard deviation (RSD%) values ≤ 30%, supporting the robustness and reproducibility of our analytical workflow. The RSD% values for all annotated metabolites are provided in the Appendix A, along with additional annotation-related information such as adduct type, molecular formula, InChIKey, total identification score, signal-to-noise ratio, MS1 isotopic pattern, and MS/MS spectrum. Among the most abundant plasma metabolites, characterized by high signal-to-noise ratios, we identified compounds such as LysoPC(18:1(11Z)), Arg-Arg-Arg, alloisoleucine, 11-hydroxyprogesterone 11-glucuronide, LysoPC(0:0/18:0), betaine, and creatine (Appendix A). AAs, peptides, and analogues were the most enriched class (327 compounds), followed by glycerophosphocholines (133 compounds) and glycerophosphoethanolamines (113 compounds). Interestingly, several lipid-like molecules, including ceramides and phosphosphingolipids, were also detected (Figure 1).

Therefore, the untargeted UHPLC-HRMS metabolomic approach proved particularly effective in the comprehensive detection of AAs, small peptides, and phospholipids. Regarding mycotoxin-derived metabolites, the untargeted approach enabled the annotation of α- and β-ZEL as well as nivalenol (Appendix A). No other plasma metabolites related to ZEN or DON were detected. In the subsequent step, both unsupervised and supervised algorithms were applied to discriminate among groups based on the plasma metabolome. In particular, the averaged hierarchical clustering analysis (HCA) shown in Figure 2A clearly revealed a more distinct metabolomic profile for the MTX group, whereas CTR and MDP samples clustered together.

Additionally, specific clusters of metabolites were found to be exclusively up- or down-accumulated in some groups. This broad chemical coverage, along with distinct accumulation trends, supports the utility of metabolomics to further explore the systemic effects of dietary mycotoxins and the mitigating role of MDP. Similarly, supervised partial least squares discriminant analysis (PLS-DA) revealed a gradient-like separation (Figure 2B), with the MDP group positioned between CTR and MTX. However, the predictive model was not statistically significant, as indicated by a Q^2^(cum) value below 0.5 (not significant).

### 2.2. Pairwise Comparisons from OPLS-DA and Venn Diagram

Considering the lack of predictive power in the initial PLS-DA model, we performed three additional pairwise OPLS-DA comparisons, namely “MDP vs. MTX”, “MDP vs. CTR”, and “MTX vs. CTR”, which revealed 262, 278, and 269 VIP-discriminant plasma metabolites, respectively (Appendix A). Among these models, the highest Q^2^(cum) value was observed for the “MDP vs. CTR” comparison (0.392), although still below the commonly accepted threshold for robust prediction (Q^2^ > 0.50). In the “MTX vs. CTR” comparison, three metabolites showed high predictive performance: chitobiose (VIP score: 2.670; log_2_FC: 0.34; AUC: 0.81; *p* < 0.05), guanidinosuccinic acid (VIP score: 2.136; log_2_FC: −0.42; AUC: 0.79; *p* < 0.05), and N(6)-Methyllysine (VIP score: 2.707; log_2_FC: −0.72; AUC: 0.79; *p* < 0.05). Additionally, 50 plasma metabolites exhibited AUC values between 0.70 and 0.80, identifying them as further candidates for evaluating the impact of mycotoxins on the plasma metabolome of dairy cows. Regarding the “MDP vs. MTX” comparison, four metabolites showed good robustness: N-lauroylglycine (VIP score: 2.315; log_2_FC: 3.65; AUC: 0.79; *p* < 0.05), aminomalonic acid (VIP score: 2.712; log_2_FC: 0.46; AUC: 0.77; *p* < 0.05), N-acetylhistidine (VIP score: 2.690; log_2_FC: 0.36; AUC: 0.77; *p* < 0.05), riboflavin (VIP score: 2.665; log_2_FC: 0.73; AUC: 0.77; *p* < 0.05), and 8-Oxo-dGMP (VIP score: 2.719; log_2_FC: −1.05; AUC: 0.77; *p* < 0.05). An additional 35 metabolites presented AUC values above 0.70 (Appendix A). The “MDP vs. CTR” comparison revealed four plasma metabolites with excellent discrimination (AUC > 0.80): 1,7-dimethylguanosine (VIP score: 3.051; log_2_FC: 0.58; AUC: 0.85; *p* < 0.05), N-undecanoylglycine (VIP score: 2.496; log_2_FC: 0.49; AUC: 0.80; *p* < 0.05), N-acetylhistidine (VIP score: 2.354; log_2_FC: 0.32; AUC: 0.80; *p* < 0.05), and homogentisic acid (VIP score: 2.187; log_2_FC: 0.39; AUC: 0.81; *p* < 0.05), In total, 68 additional metabolites from this comparison exhibited AUC values greater than 0.70 (Appendix A). All discriminant metabolites were used to generate a Venn diagram (Figure 3), highlighting unique and shared metabolomic signatures. Specifically, 84 metabolites were exclusively altered in the “MDP vs. MTX” comparison, while 93 were specific to “MDP vs. CTR”, and 83 were exclusive to “MTX vs. CTR”.

A full list of exclusive and overlapping plasma metabolites is provided in the Appendix A. Of particular interest, 81 metabolites were identified as exclusively associated with MDP supplementation, potentially representing its plasma metabolomic signature. Among these, 9 compounds showed consistent log_2_FC values (indicative of accumulation) and robust classification performance based on receiver-operating characteristic curve (ROC) analysis (AUC values ≥ 0.7). However, it should be noted that some of these differences may also reflect additional effects related to the specific plant- and algae-derived components in the MDP formulation, beyond direct detoxification of ZEN and DON. These 9 discriminant plasma metabolites are reported in Table 1. 

### 2.3. Pathway Analyses and Exclusive Biomarker Compounds of MDP

Pathway analyses were performed using the *B. taurus* metabolome (KEGG) as a reference and based on the VIP-discriminant metabolites identified in the different pairwise comparisons. The results are presented in Figure 4A–C for the “MDP vs. MTX”, “MDP vs. CTR”, and “MTX vs. CTR” comparisons, respectively.

In the “MDP vs. MTX” comparison, the most significantly affected pathways were tryptophan metabolism, linoleic acid metabolism, sphingolipid metabolism, and glycerophospholipid metabolism (Figure 4A). When comparing MDP with CTR, two main pathways emerged as significantly impacted: glycerophospholipid metabolism and pyrimidine metabolism (Figure 4B). Regarding the “MTX vs. CTR” comparison, the most prominent alterations involved sphingolipid metabolism and steroid hormone biosynthesis (Figure 4C). A comprehensive overview of the number of VIP compounds associated with each metabolic pathway, their cumulative log_2_FC values, and the most discriminant and robust VIP compound per pathway is provided in Table 2.

Overall, glycerophospholipids were markedly down-accumulated in the MDP group, with cumulative log_2_FC values of −17.46 and −24.47 when compared to MTX and CTR, respectively. Interestingly, sphingolipids were globally down-accumulated in the MDP group (log_2_FC: −1.31) compared to MTX, indicating a potential shift in sphingolipid remodeling. In the “MTX vs. CTR” comparison, sphingolipids were globally up-accumulated under MTX exposure, with ceramide (t18:0/16:0) identified as the most discriminant metabolite. Additionally, steroid hormone biosynthesis was also affected in the MTX group, with pregnanetriol emerging as the most discriminant and significant compound in this pathway. Remarkably, the untargeted metabolomic approach also enabled the putative annotation of 8-oxo-dGMP, a marker of oxidative DNA damage (Appendix A). This metabolite was significantly down-accumulated in the MDP group compared to MTX, suggesting a potential protective effect of the detoxifying product on oxidative stress. All additional metabolites assigned to the various metabolic pathways, along with their corresponding log_2_FC and AUC values, are available in the Appendix A.

## 3. Discussion

### 3.1. Metabolic Disruptions Caused by Fusarium Mycotoxins (MTX vs. CTR)

In this work, we investigated the systemic metabolomic alterations associated with feeding *Fusarium*-contaminated diets to early-lactation dairy cows and examined the potential of a commercial mycotoxin-deactivating product (MDP) to modulate these metabolomic responses, using an untargeted plasma metabolomics approach. To date, few studies have investigated the systemic metabolic effects of *Fusarium* mycotoxins in dairy cows using plasma metabolomics. Most previous works focused on the presence of mycotoxin metabolites in biofluids [2,15]. Only a limited number of studies applied metabolomic approaches; particularly, Dong et al. [11] reported DON-induced changes in rumen fluid metabolites, while Wang et al. [13] observed alterations in plasma amino acid metabolism using ^1^H NMR. Ogunade et al. [16] investigated plasma metabolomics in response to aflatoxin B1. Overall, comprehensive plasma metabolomics studies assessing the impact of *Fusarium* toxins remain scarce, highlighting the novelty of the present work and its complementarity to the previous study by Catellani et al. [14].

Building on this background, we next describe the specific plasma metabolomic alterations observed in cows exposed to *Fusarium*-contaminated diets (MTX) compared with control animals (CTR), focusing on key metabolic pathways affected by mycotoxin intake. The pairwise comparison revealed several metabolomic perturbations, particularly involving sphingolipid metabolism, steroid hormone biosynthesis, and several biomarkers of oxidative stress and endocrine interference. Within the sphingolipid pathway, several complex sphingolipids and bioactive intermediates were up-accumulated in MTX cows, including ceramide (t18:0/16:0) (log_2_FC: 0.80; AUC: 0.69), lactosylceramide (d18:1/12:0) (log_2_FC: 0.67; AUC: 0.71), and trihexosylceramide (d18:1/12:0) (log_2_FC: 0.60; AUC: 0.60). These compounds are involved in cell signaling, membrane integrity, and inflammation [17], and their accumulation is consistent with the known inhibition of ceramide synthase by FUM [18,19]. Although no additional FUM was added in this trial, and the dietary challenge was focused on ZEN and DON, the accumulation trend of sphinganine 1-phosphate (log_2_FC: 0.44; AUC: 0.71; *p* > 0.05) and other sphingolipids suggests that DON and ZEN alone can affect sphingolipid homeostasis, possibly through indirect modulation of ceramide metabolism or interference with membrane remodeling processes. This finding aligns with previous literature reports, indicating that DON and ZEN may trigger events related to lipid metabolism, especially events connecting the digestive system and lipid metabolism, that potentially accelerate the toxic effect of FUM on membrane lipids [20,21]. Although not significant from Volcano plot analysis, the simultaneous up-accumulation of sphingomyelins, including the VIP compounds SM(d16:0/18:2), SM(d17:0/24:1), and SM(d18:1/22:2(OH)), further suggests a potential activation of stress-related membrane remodeling and potential endoplasmic reticulum stress [22]. These lipids participate in membrane microdomain structure and may reflect compensatory stabilization under toxic pressure. The increased abundance of sphinganine 1-phosphate reinforces this mechanism, as it is a well-documented marker of disrupted ceramide biosynthesis. In fact, even without direct fumonisin exposure, elevated sphinganine (Sa) and sphingosine (So) may arise as a secondary consequence of DON/ZEN-induced stress, as both compounds can be influenced by inflammatory cytokine signaling and oxidative imbalance. It is worth noting that, in a previous publication from the same trial [14], no significant differences were observed in standard blood biochemistry parameters between the groups, and no clear diet effect was found for plasma anti-Müllerian hormone (AMH). Similarly, the present untargeted metabolomics analysis, which was conducted on plasma collected at 56 DIM without parity stratification, outlined scarcely significant (*p* > 0.05) but consistent metabolic shifts, particularly in sphingolipid intermediates, oxidative stress markers, and steroid hormones, that are not always detectable through conventional biochemical assays. This supports the complementarity of untargeted metabolomics in capturing subclinical metabolic adaptations beyond blood chemistry parameters.

Although not significant (*p* > 0.05) when considering Volcano plot analysis, log_2_FC values observed for some VIP metabolites involved in steroid hormone biosynthesis suggested a hypothetical systemic endocrine disruption in MTX cows. The putative impact of ZEN on the steroidogenic pathway in *B. taurus* is illustrated in Figure 5. The untargeted plasma metabolomics revealed up-accumulation trends (*p* > 0.05) for estradiol (log_2_FC: 0.46; AUC: 0.65), estrone (log_2_FC: 0.41; AUC: 0.63), and 17β-estradiol 3-sulfate-17-(β-D-glucuronide), suggesting enhanced estrogenic stimulation, consistent with the known estrogen-mimicking activity of ZEN binding to estrogen receptors and altering hypothalamic-pituitary-gonadal axis feedback [23]. Additionally, the up-accumulation trends of cholesterol sulfate (log_2_FC: 1.04; AUC: 0.63) and 17-hydroxypregnenolone sulfate (log_2_FC: 0.28; AUC: 0.65) may reflect intensified steroidogenic activity upstream and enhanced sulfation as a regulatory mechanism to reduce excess free steroid hormones, rather than a simple downstream block. Notably, several androgenic and progestin metabolites were down-accumulated in MTX cows compared to CTR, including pregnanetriol (log_2_FC: –0.48; AUC: 0.68; *p* < 0.05), androsterone, etiocholanolone glucuronide, tetrahydrodeoxycorticosterone, and 19-norandrosterone 3-glucuronide. This pattern may indicate a functional suppression of luteal and adrenal steroidogenesis, likely related to the oxidative stress imposed by DON [24] and the disruption of membrane lipid signaling induced by FB1, which together compromise the secretory capacity of steroidogenic tissues. The increase of 11-oxo-androsterone glucuronide (log_2_FC: 0.72) may reflect a shift in androgen metabolism and hepatic clearance, resulting in accumulation of glucuronidated forms.

This multi-steroid metabolic fingerprint aligns with the physiological trends previously described in the same animals [14]. Specifically, the earlier study reported that cows fed *Fusarium*-contaminated diets showed a slower increase in milk progesterone concentrations, fewer corpora lutea (CLs) detected by ultrasound, and a higher proportion of postpartum anestrus cows, especially among primiparous animals. These physiological alterations are compatible with delayed resumption of cyclicity, as confirmed by survival analysis, and a higher tendency for the first dominant follicle (F1) to regress or become cystic rather than ovulate, a pattern that parallels the accumulation of ZEN-derived α-ZEL and β-ZEL metabolites interfering with key steroidogenic enzymes (3α/3β-HSD), which play pivotal roles in progesterone and androgen synthesis [25,26]. By linking these physiological observations to specific metabolic markers, the present plasma metabolomics analysis extends the previous findings by providing molecular-level evidence that dietary *Fusarium* toxins not only mimic estrogenic effects but also shift the balance of steroid biosynthesis, clearance, and conjugation. The discriminant accumulation of conjugated estrogen metabolites, together with the decrease in progestins and active androgens, supports the notion that these mycotoxins can alter the delicate balance between estrogenic stimulation and luteal function, particularly during early lactation when cows are under metabolic stress. Taken together, these results indicate that the untargeted metabolomic approach can uncover metabolic patterns potentially related to subtle endocrine alterations, which may be consistent with the physiological delay in cyclicity previously observed. This approach, therefore, contributes to linking conventional reproductive monitoring with molecular-level metabolic insights. Additionally, the significant up-accumulation of 8-oxo-dGMP (log_2_FC: 0.54; *p* < 0.05), a metabolite previously associated with oxidative DNA damage [27], suggests a possible increase in oxidative stress under mycotoxin exposure. In the same trial [14], higher ROMt concentrations were observed only in MTX-fed primiparous cows, indicating a parity-dependent oxidative response. The present untargeted metabolomics analysis complements those findings by showing that, at 56 DIM and across the full cohort, the MTX group exhibited elevated plasma levels of 8-oxo-dGMP, which may reflect enhanced oxidative processes at the molecular level. Taken together, these observations point toward a consistent oxidative challenge in cows exposed to moderate *Fusarium* contamination and underline the usefulness of combining conventional biochemical assays with metabolomics for a more comprehensive assessment of metabolic stress.

### 3.2. Protective Effects of the Mycotoxin-Deactivating Product (MDP vs. MTX)

Dietary supplementation with the MDP reversed several of the disruptions observed in MTX cows. Particularly, pathway analysis highlighted significant modulation of tryptophan metabolism, sphingolipid metabolism, and glycerophospholipid metabolism, with consistent effects on specific metabolite levels. Within sphingolipid metabolism, MDP-fed cows showed a significant (*p* < 0.05) down-accumulation of sphinganine 1-phosphate (log_2_FC: −1.40; AUC: 0.68) and lactosylceramide (d18:1/12:0) (log_2_FC: −0.37; AUC: 0.64), suggesting recovery of ceramide synthase activity and reduced inflammatory lipid signaling. Although not significant from Volcano plot analysis, the accompanying down-accumulation of sphingomyelins, such as SM(d18:1/16:1(9Z)(OH)) and SM(d18:0/16:0), supports a reduction in membrane-derived inflammatory signaling and indicates improved membrane homeostasis. Also, the significant increase (*p* < 0.05) in L-serine in MDP suggests improved precursor availability for normalized sphingolipid flux. In the tryptophan pathway, several metabolites were increased, namely N-acetylserotonin (log_2_FC: 0.35; AUC: 0.69; *p* < 0.05), 4,6-dihydroxyquinoline (log_2_FC: 0.29; AUC: 0.74; *p* < 0.05), formylanthranilic acid (log_2_FC: 0.29; AUC: 0.65; *p* > 0.05), and indole-acetaldehyde (log_2_FC: 0.48; AUC: 0.63; *p* > 0.05). These changes may reflect enhanced kynurenine flux under detoxified conditions. Also, 2-aminomuconic acid (log_2_FC: −0.37; *p* > 0.05) and indole-3-acetic-acid-*O*-glucuronide (log_2_FC: −0.33; *p* < 0.05) were both down-accumulated. Tryptophan (Trp) is an essential aromatic AA previously linked to the regulation of inflammatory responses in dairy cows with ketosis [28]. However, the association between Trp metabolism and inflammation in dairy cows fed a mycotoxin diet remains unclear. Trp metabolism follows three major pathways: (1) Trp is converted into kynurenine (KYN) in the liver and immune tissue; (2) serotonin or 5-hydroxytryptamine is generated from Trp in enterochromaffin cells via hydroxylation, and the serotonin is acetylated to form N-acetylserotonin to finally produce melatonin; (3) Trp is directly transformed into several molecules, such as indole-derivatives by the gastrointestinal microbiota [28]. Serotonin and its derivatives are involved in the regulation of metabolic homeostasis in dairy cows during transition periods [29]. The role of serotonin in calcium metabolism, mammary gland function, and animal welfare in dairy cows is increasingly recognized [30]. This compound acts as a key regulator of calcium homeostasis by influencing both intestinal absorption and bone mobilization. In the mammary gland, it modulates milk synthesis and tight junction integrity, contributing to udder health [30]. Serotonin has also been linked to behavioral responses and stress adaptation, suggesting a broader role in supporting animal welfare during critical phases such as early lactation. Additionally, the significantly up-accumulated N-acetylserotonin (also known as normelatonin) has been previously linked to antioxidant and anti-inflammatory effects [31] and may reflect improved systemic resilience. However, given the pleiotropic nature of N-acetylserotonin and the scarce evidence available in ruminants, this interpretation should be considered exploratory, as its antioxidant or anti-inflammatory roles have been mostly described in monogastric models or cellular systems. Interestingly, the concurrent significant reduction in indole-3-acetic-acid-*O*-glucuronide (log_2_FC: −0.33) may reflect a shift in tryptophan catabolism away from conjugated excretion and toward more functionally active pathways. This metabolite represents an inactivated and water-soluble form of indole-3-acetic acid, typically destined for hepatic clearance via glucuronidation. This down-accumulation suggests a reduced flux toward elimination and a potential redirection of tryptophan-derived intermediates toward regulatory routes, such as serotonin and kynurenine metabolism. The alteration in glycerophospholipid metabolism observed in MDP-supplemented cows suggests selective remodeling rather than a generalized suppression of lipid synthesis. The significant (*p* < 0.05) down-accumulation of specific PC species, such as PC(18:1/22:4), may reflect reduced availability of polyunsaturated phospholipids for inflammatory eicosanoid production, while the concurrent up-accumulation of other structural PC and PG species (e.g., PC(20:4/18:0), PG(18:1/16:1)) supports a remodeling process that favors membrane stabilization and integrity [32]. These patterns are coherent with MDP modes of action (Mycofix^®^ Plus, BIOMIN Holding GmbH, Tulln, Austria), namely adsorption, biotransformation, and bioprotection at the gastrointestinal level. Additionally, once absorbed, its plant- and algae-derived bioactives may contribute to systemic metabolic modulation. Altogether, the MDP-fed cows displayed a metabolomic phenotype characterized by lipid stabilization, oxidative repair, and neuroimmune modulation, highlighting the complementary roles of local detoxification and bioactive-driven metabolic regulation.

Finally, among the most discriminant metabolites identified in Table 1, which collectively outline the metabolic fingerprint of MDP supplementation in dairy cow plasma, the previous literature provides partial support for their biological relevance. Luo et al. [33] identified pipecolic acid as a significant plasma biomarker around parturition in dairy cows using untargeted metabolomics, suggesting a link with metabolic adaptation during physiologically demanding stages. Similarly, riboflavin (vitamin B2), highlighted in our dataset, has been reported as a biomarker in early-lactation cows supplemented with rumen-protected glucose [34]. Riboflavin plays a central role in mitochondrial energy production and in amino acid, carbohydrate, and lipid metabolism, and contributes to oxidative stress alleviation through its function as a cofactor for flavin-dependent enzymes, thereby protecting tissues from lipid peroxidation. However, given the multifaceted physiological roles of these metabolites and the limited evidence available under mycotoxin-challenge conditions, their interpretation in the context of MDP supplementation should be regarded as exploratory and hypothesis-generating.

### 3.3. Exploratory Metabolomic Differences Between MDP and CTR Groups

The metabolomic comparison between MDP and CTR groups revealed subtle systemic metabolic shifts in cows receiving the MDP, even though the primary objective of the study was to evaluate its efficacy under *Fusarium* contamination rather than its intrinsic metabolic effects. Because the MDP group was fed a moderately contaminated diet supplemented with the product, while CTR cows received only a mildly contaminated diet, direct comparison between the two groups must be interpreted with caution, as differences in mycotoxin exposure could partly account for the observed metabolic variations. In order to clarify the structure of the experiment, it is important to note that the present study was conducted on the same animals, diets, and experimental design previously described in the companion paper by Catellani et al. [14]. That trial was intentionally conceived to evaluate whether the MDP could mitigate the systemic effects of *Fusarium* toxins under practical farm-level contamination scenarios. For this reason, the MDP was tested exclusively under the highest contamination level, and no additional groups, such as a toxin-free control or a toxin-free diet supplemented with MDP, were included. The design, therefore, reflects an applied objective of assessing protective efficacy rather than the stand-alone metabolic activity of the additive. As a result, any differences observed between MDP and CTR should be regarded as exploratory and influenced, at least in part, by the differing mycotoxin loads rather than by direct effects of the product alone. The MDP used in this trial includes bentonite, the enzyme ZenA, microbial-biological components, and a plant- and algae-derived bioprotection mix. According to the available literature [35], these phycophytic compounds support immune and liver function and reinforce the intestinal barrier in the presence of mycotoxins. In line with this, we did not expect major metabolomic effects in cows hypothetically receiving MDP alone, but rather a normalization of plasma metabolites disrupted by *Fusarium* toxins. The untargeted metabolomic data suggest that MDP supplementation may have influenced several metabolic routes beyond direct detoxification. The marked down-accumulation of glycerophospholipids (cumulative log_2_FC: −24.47), including phosphatidylethanolamines and phosphatidylcholines, may reflect adjustments in membrane lipid turnover and cellular energy balance. Alterations in the pyrimidine pathway, such as increased CDP and orotidine and decreased uridine 5′-monophosphate, could similarly indicate modulation of nucleotide metabolism and cellular maintenance processes. Furthermore, exclusive metabolites such as homogentisic acid, allantoic acid, and 1,7-dimethylguanosine point toward regulated oxidative and nitrogen metabolism. However, given the absence of a specific “CTR + MDP” treatment, it is not possible to determine whether these metabolomic shifts arise from the bioactive compounds present in the MDP (e.g., plant and algae extracts) or from differences in mycotoxin load between diets. Future studies will be required to evaluate the direct metabolic influence of the product under toxin-free conditions. As detailed in the companion paper [14], the experimental design was intentionally structured to evaluate whether the product could mitigate the systemic consequences of *Fusarium* mycotoxins in early-lactation dairy cows. The ZEN (55.42 µg/kg DM) and DON (226.8 µg/kg DM) levels in the CTR diet were well below the European Commission Recommendation 2006/576/EC guidance limits (500–3000 µg/kg for ZEN; 5000 µg/kg for DON), yet representative of realistic on-farm scenarios. In this context, the comparison between MDP and CTR helps determine whether MDP partially normalizes the metabolomic phenotype under practical feeding conditions. Nevertheless, such comparisons must be interpreted within the study’s primary aim, which focused on protective efficacy rather than on additive-induced metabolic modulation.

## 4. Conclusions

This study provides an exploratory overview of the systemic metabolic alterations associated with exposure to *Fusarium* mycotoxins in early-lactation dairy cows and the potential mitigating effects of a mycotoxin-deactivating product (MDP). The untargeted metabolomic analysis revealed distinct alterations in lipid, nucleotide, and redox-related pathways in cows fed the contaminated diet, suggesting subclinical impacts on metabolic homeostasis despite the absence of overt clinical or productivity impairments previously reported for this trial. Supplementation with the MDP partially modulated several of these metabolite classes, particularly glycerophospholipids and pyrimidine intermediates, indicating a possible normalization of metabolic processes disrupted by mycotoxin exposure. However, these findings rely exclusively on metabolomic endpoints and should therefore be interpreted as hypothesis-generating. The absence of a toxin-free group supplemented with MDP limits the ability to disentangle the direct effects of the product from interactions with mycotoxin load. Moreover, the cross-sectional design does not allow assessment of temporal dynamics. Overall, the results suggest that metabolomics can detect subtle, systemic signatures of *Fusarium* mycotoxin exposure in dairy cows and indicate a potential protective influence of the MDP under field-relevant contamination levels. Future studies incorporating longitudinal sampling, additional physiological and clinical endpoints, and an MDP-only treatment arm will be necessary to confirm the biological significance and mechanistic basis of these preliminary observations.

## 5. Materials and Methods

### 5.1. Experimental Cows and Diets

Animal experimentation was in compliance with DPR 27/1/1992 (Animal Protection Regulations of Italy) in conformity with European Community regulation 86/609 and was authorized by the Italian Ministry of Health in accordance with Italian Health regulations (n° 112/2022-PR). This study was carried out at the experimental dairy farm of the Università Cattolica del Sacro Cuore di Piacenza (CERZOO, San Bonico, Piacenza, Italy). All the details regarding the Holstein dairy cows enrolled, together with ingredients (%DM) and chemical composition (%Dry matter, DM) of the diets fed to lactating Holstein cows from calving to 56 days in milk (DIM), are fully reported in Catellani et al. [14]. Thirty Holstein cows were selected according to parity (9 primiparous and 21 multiparous; 3.47 ± 1.17 previous lactations) and BW at calving (683.24 ± 36.78 kg). All animals were housed in the same freestall barn equipped with cubicles and a shared feeding and resting area. Cows were milked twice daily (0430 and 1630 h) and had free access to water. Before treatment allocation, cows were blocked by BW (primiparous vs. multiparous), parity order within multiparous animals, and previous lactation yield, and subsequently assigned at random to one of three experimental groups (3 primiparous and 7 multiparous per group). Body condition scoring (dimensionless) was weekly performed by the same operator using a 1 to 5 scale after the morning feed, being comparable among dietary groups and not affected by parity [14]. The experimental groups were the following: animals fed with a total mixed ration (TMR) that was (1) mildly contaminated with *Fusarium* mycotoxins (Control group, CTR; n = 10; ZEN = 55.42 μg/kg DM and DON = 226.8 μg/kg DM); (2) moderately contaminated with *Fusarium* mycotoxins (Mycotoxin group, MTX; n = 10; ZEN = 366.63 μg/kg DM, being 6-fold higher than CTR group, and DON = 1141.54 μg/kg DM, being 5-fold higher than CTR group); and (3) moderately contaminated with *Fusarium* mycotoxins but added with the mycotoxin-deactivating product (Mycofix^®^ Plus, BIOMIN Holding, GmbH, Tulln, Austria) (Mycotoxin-deactivating product group, MDP; n = 10; ZEN = 319.72 μg/kg DM, being 5.7-fold higher than CTR group, and DON = 1028.42 μg/kg DM, being 4.5-fold higher than CTR group). The MDP tested in current trial consisted of four main components: (1) an inorganic material (i.e., bentonite); (2) the enzyme zearalenone hydrolase ZenA (ZENzyme^®^), which converts ZEN to hydrolyzed ZEN (HZEN), by hydrolyzing the ester bond of its lactone ring [36]; (3) microbial and biological constituents; (4) a bioprotection mix, consisting in a blend of extracts from plants and algae [37,38]. Furthermore, FUM (as the sum of FB1 and FB2) showed similar values for all experimental groups, being 892.39 μg/kg DM, 951.55 μg/kg DM, and 842.44 μg/kg DM in CTR, MTX, and MDP, respectively. As described in Catellani et al. [14], mycotoxin concentrations in the experimental diets were monitored weekly throughout the 8-week feeding period to ensure stability of contamination levels. Mycotoxin quantification followed the chromatographic procedures reported by Pietri and Bertuzzi [39] for fumonisins (FUM) and by Bertuzzi et al. [40] for zearalenone (ZEN) and deoxynivalenol (DON). Although multi-mycotoxin LC–MS/MS methods are currently available for the simultaneous determination of *Fusarium* mycotoxins, in the present long-term feeding trial ZEN and FUM were quantified by HPLC-FLD and DON by GC-MS in order to maintain analytical continuity with an already validated workflow routinely applied in our laboratory, as well as with the companion study by Catellani et al. [14], thus ensuring full comparability across experimental groups. Briefly, FUM was extracted using a phosphate buffer, purified through an immuno-affinity column (R-Biopharm Rhone Ltd., Glasgow, Scotland), and quantified by HPLC-FLD. The limit of detection (LOD) and limit of quantification (LOQ) for FUM were 10 and 30 μg/kg, respectively. ZEN and DON were extracted with acetonitrile/water (86:14, *v*/*v*). ZEN was purified using an immuno-affinity column and quantified by HPLC-FLD, whereas DON was purified using a Trilogy-Puritox Trichothecenes column (R-Biopharm Rhone Ltd.) and quantified by GC–MS. The LOD and LOQ were 2 and 5 μg/kg for ZEN and 10 and 30 μg/kg for DON, respectively. All chromatographic conditions, validation parameters, and recovery rates are reported in detail in Catellani et al. [14]. Across the feeding period, ZEN ranged from 49.2 to 64.4 µg/kg DM in CTR, 351.4 to 396.8 µg/kg DM in MTX, and 317.4 to 329.6 µg/kg DM in MDP diets. DON concentrations varied between 121.1 and 349.1 µg/kg DM in CTR, 957.0 and 1181.7 µg/kg DM in MTX, and 1001.8 and 1102.7 µg/kg DM in MDP diets. FUM ranged from 845.0 to 1063.0 µg/kg DM in CTR, 878.0 to 886.4 µg/kg DM in MTX, and 736.5 to 834.6 µg/kg DM in MDP diets. The MTX ratio was supplemented with 0.19% of dry matter (DM) of mycotoxin-contaminated culture medium (CM) and 0.15% DM of placebo to achieve the targeted contamination level. The MDP TMR included 0.19% DM of CM and 0.15% DM of MDP (35 g/cow/d), whereas 0.34% DM of placebo was added to the CRT ration. Particularly, 34.6% DM corn silage and 12.6% DM sorghum silage were used in the basal diets fed to lactating Holstein cows. All the additional details related to ingredients (%) and chemical composition (%DM) of the diets are fully reported in Catellani et al. [14].

### 5.2. Collection of Plasma Samples and Extraction of Metabolites

Blood samples were collected weekly in the morning (before 8:00 feeding) from 7 to 56 DIM by venipuncture of the jugular vein using 10 mL lithium heparin tubes (Vacuette, containing 18 IU of Li-heparin/mL, Kremsmunster, Austria) and immediately cooled in an ice water bath [14]. A blood aliquot was used to calculate packed cell volume [14], whereas the remaining sample was centrifuged (3500× *g* for 16 min at 4 °C) and the resulting plasma was separated into aliquots and stored at −20 °C until analysis. An aliquot of plasma obtained from samples collected at the end of the experimental period (56 DIM) was then used for the untargeted metabolomics analysis. A total of 60 plasma samples were analyzed, corresponding to 10 samples per experimental group, each analyzed in duplicate (3 groups × 10 samples × 2 replicates). The extraction of metabolites was carried out as previously described by Cattaneo et al. [41], with some modifications. Briefly, samples were slowly thawed at 4 °C, and 200 μL of plasma was added to 800 μL of pre-cooled methanol/acetonitrile solution (1:1, *v*/*v*). The extraction step was promoted by an ultrasound-assisted extraction (UAE; DU-32 ARGOLab, Milan, Italy; power level: 120 W) at room temperature for 15 min. Thereafter, plasma samples were vortex mixed and centrifuged at 14,000× *g* at 4 °C for 20 min, followed by an overnight incubation at −20 °C. The resulting supernatants were then filtered through 0.22 μm cellulose syringe filters in ultra-HPLC vials for untargeted metabolomic profiling.

### 5.3. Untargeted Metabolomic Profiling Based on HRMS

The untargeted UHPLC-HRMS analysis was conducted using a Q-Exactive Focus Hybrid Quadrupole-Orbitrap Mass Spectrometer (Thermo Scientific, Waltham, MA, USA) coupled to a Vanquish ultra-HPLC pump, equipped with a heated electrospray ionization (HESI)-II probe (Thermo Scientific, USA). A detailed description of the instrumental conditions as related to both chromatography and mass spectrometry can be found in Cattaneo et al. [41]. Overall, the chromatographic separation was achieved using the ACQUITY UPLC Waters BEH C18 column (2.1 × 100 mm, 1.7 µm), considering a gradient elution (6–94% acetonitrile in 35 min) and 0.1% formic acid as phase modifier. The flow rate was 200 μL/min, injecting 6 μL of each sample. The untargeted analysis was conducted in the full scan range of 80–1200 m/z, using a positive ionization mode and setting a mass resolution of 70,000. Additionally, a pooled quality control sample was randomly injected and analyzed in a data-dependent (Top N = 3, under a stepped normalized collisional energy) MS/MS mode, considering a full-scan mass resolution of 17,500. The HESI parameters together with automatic gain control values and maximum injection times are reported elsewhere [41]. The collected raw data were further processed using the software MS-DIAL (version 4.90) for the automatic peak finding, LOWESS normalization, and annotation via spectral matching against the comprehensive Bovine Metabolome Database [42] (last accessed date: June 2025). The identification step was based on mass accuracy, isotopic pattern, and spectral matching, thus reaching a level 2 of confidence in annotation [43].

### 5.4. Multivariate Statistics and Pathway Analysis

The metabolomics-based dataset was further elaborated for multivariate statistical modeling using the software MetaboAnalyst 6.0 [44]. The annotated metabolites were filtered by standard deviation (percentage to filter out: 40%), normalized by median, log_10_-transformed, and Pareto scaled. After normalization, both unsupervised and supervised multivariate statistics were carried out. The unsupervised approach was based on hierarchical cluster analysis, while the partial least squares discriminant analysis (PLS-DA) was considered as a supervised tool. The PLS-DA model validation parameters (goodness-of-fit R^2^Y together with goodness-of-prediction Q^2^Y) were recorded, and the model was checked for overfitting by permutation testing (N  >  100). The discriminant potential of each plasma metabolite was then calculated according to the variable selection method variable importance in projection (VIP), using as the minimum significant threshold a VIP score  ≥  1. Volcano plot analysis, combining a Fold-Change analysis (FC cut-off > 1.2) and ANOVA (*p* < 0.05), was performed for each pairwise comparison, namely “MDP vs. MTX”, “MDP vs. CTR”, and “MTX vs. CTR”. Furthermore, to validate the VIP markers proposed for each condition, the Receiver Operating Characteristics (ROC) curves were extrapolated using the software MetaboAnalyst 6.0. To this aim, the area under the ROC curve (AUC) was inspected to evaluate the global performance of each discriminant marker proposed. The exclusive VIP metabolites of each pairwise comparison were selected through Venn’s analysis in the Venny 2.1.0 online software (https://bioinfogp.cnb.csic.es/tools/venny/, accessed on 1 June 2025). Finally, the software MetaboAnalyst 6.0 was used to inspect those metabolic pathways mostly represented by the discriminant metabolites annotated, using as pathway library *Bos taurus* (Kyoto Encyclopedia of Genes and Genomes, KEGG).

## Figures and Tables

**Figure 1 toxins-18-00009-f001:**
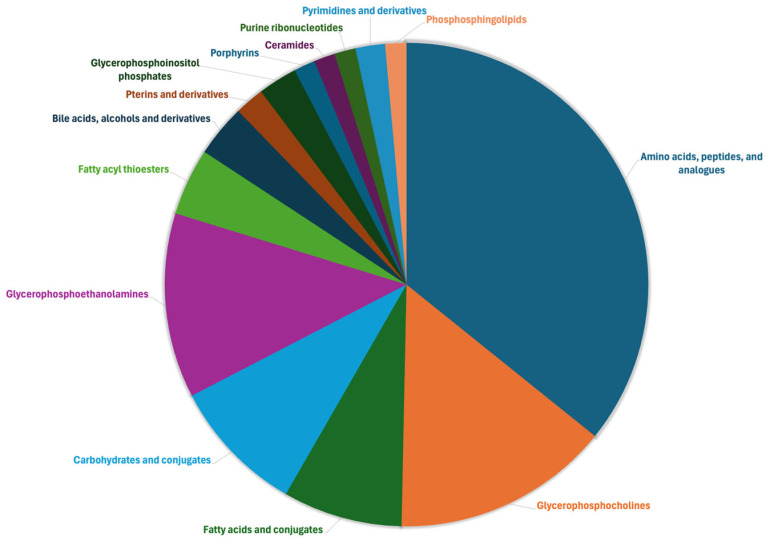
Pie chart showing the chemical classes annotated by untargeted metabolomics in the different plasma samples, considering the classes of metabolites containing at least two entries.

**Figure 2 toxins-18-00009-f002:**
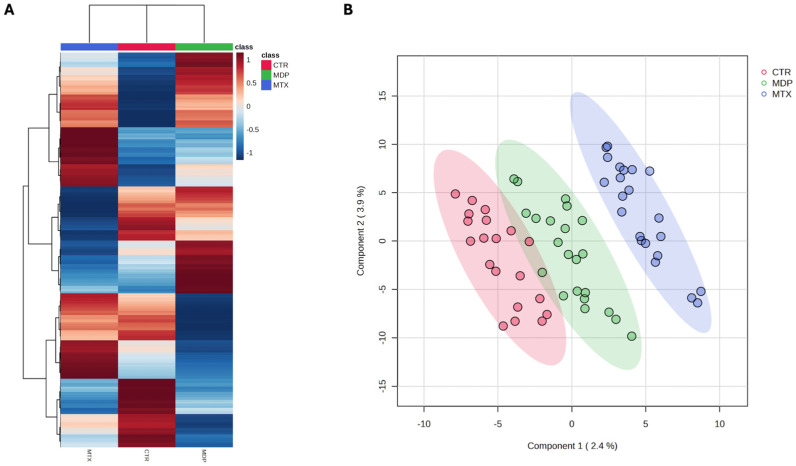
(**A**) Averaged heat map resulting from unsupervised hierarchical cluster analysis; (**B**) Supervised PLS-DA score plot showing the separation of the different groups according to the plasma metabolomic profiles. Goodness of fitting (R^2^Y) = 0.940; Goodness of prediction (Q^2^) = not significant.

**Figure 3 toxins-18-00009-f003:**
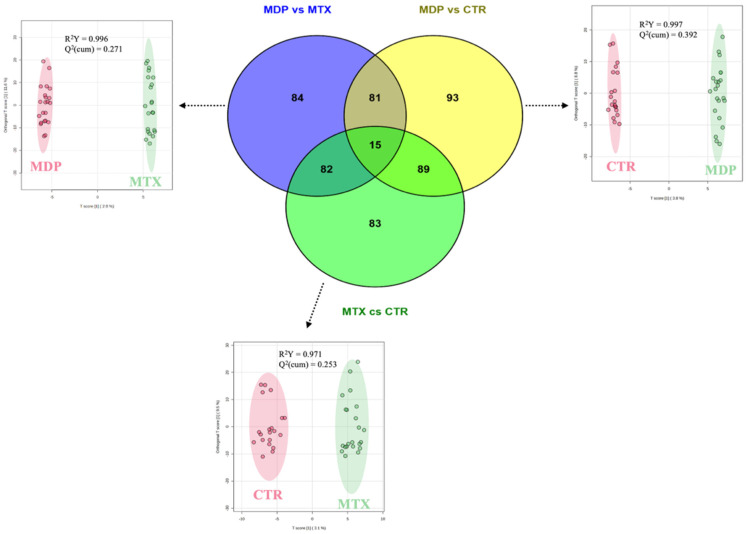
Venn diagram considering the VIP discriminant metabolites of each OPLS-DA model under investigation for the pairwise comparisons “MDP vs. MTX”, “MDP vs. CTR”, and “MTX vs. CTR”.

**Figure 4 toxins-18-00009-f004:**
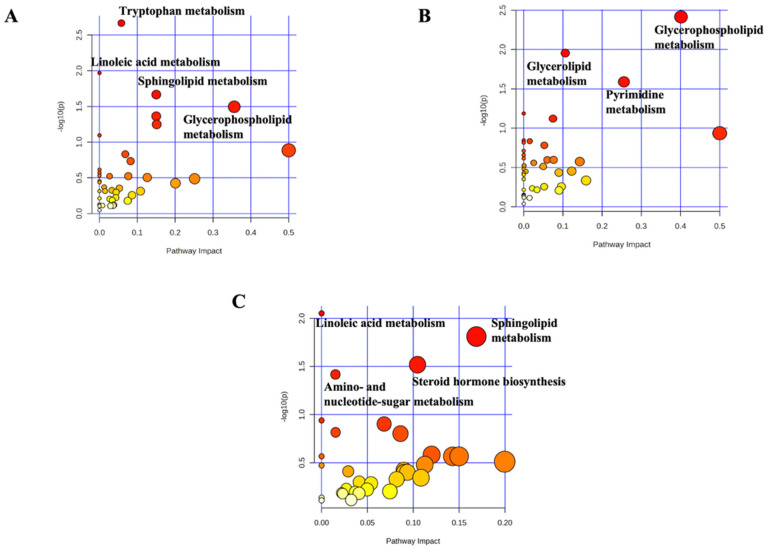
Pathway analyses considering the *B. taurus* metabolome and the pairwise comparisons. (**A**) = MDP vs. MTX; (**B**) = MDP vs. CTR; (**C**) = MTX vs. CTR.

**Figure 5 toxins-18-00009-f005:**
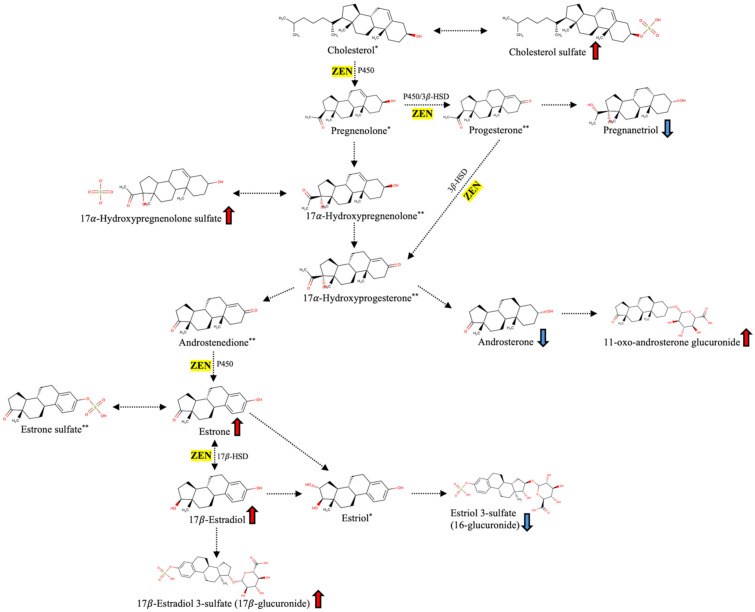
Putative impact of ZEN and its metabolites on steroid hormone biosynthesis in dairy cows (*Bos taurus*). ZEN disrupts the normal steroidogenic pathway by inhibiting key enzymatic steps, including cytochrome P450 enzymes, 3β-hydroxysteroid dehydrogenase (3β-HSD), and 17β-hydroxysteroid dehydrogenase (17β-HSD). These disruptions impair the conversion of pregnenolone to progesterone and androgens, ultimately leading to altered estrogen synthesis. α-ZEL and β-ZEL further stimulate aromatase activity and directly bind to estrogen receptors (ER), mimicking endogenous estrogens and promoting a hyperestrogenic state. * = metabolite not annotated by UHPLC-HRMS; ** = metabolite with low predictive value (VIP score < 1). Arrow colors represent metabolite accumulation in MTX vs. CTR: red indicates up-accumulation; blue indicates down-accumulation.

**Table 1 toxins-18-00009-t001:** Discriminant and common VIP metabolites representing the fingerprint of MDP in the plasma of dairy cows. Each compound is reported with its log_2_ Fold-Change (FC) value, *p*-value, and AUC according to the ROC analysis (AUC in the range 0.7–0.8 represents a fair biomarker).

Discriminant Marker of MDP	MDP vs. MTX			MDP vs. CTR		
	Log_2_ FC	AUC (ROC)	*p*-Value	Log_2_ FC	AUC (ROC)	*p*-Value
N-Acetylhistidine	0.36	0.77	2.6 × 10^−3^	0.32	0.80	2.4 × 10^−3^
Aminomalonic acid	0.46	0.77	2.2 × 10^−3^	0.46	0.79	1.6 × 10^−3^
8-Hydroxyadenine	0.32	0.74	1.5 × 10^−2^	0.37	0.77	7.8 × 10^−3^
Pipecolic acid	0.30	0.71	1.1 × 10^−2^	0.34	0.74	5.5 × 10^−3^
PC(20:1(11Z)/18:2(9Z,12Z))	−0.59	0.77	>0.05	−0.57	0.75	4.1 × 10^−2^
2-(Formamido)-N1-(5-phospho-D-ribosyl)acetamidine	0.45	0.72	7.9 × 10^−3^	0.49	0.74	5.3 × 10^−3^
Allantoic acid	0.46	0.74	6.2 × 10^−3^	0.43	0.73	1.1 × 10^−2^
Riboflavin	0.73	0.77	2.9 × 10^−3^	0.59	0.73	1.5 × 10^−2^
4-(2-Amino-3-hydroxyphenyl)-2,4-dioxobutanoic acid	0.38	0.71	2.4 × 10^−2^	0.36	0.72	4.1 × 10^−2^

**Table 2 toxins-18-00009-t002:** Main metabolic pathways characterizing the different pairwise comparisons under investigation, namely MDP vs. MTX, MDP vs. CTR, and MTX vs. CTR. Each pathway is reported together with the total number of variable importance in projections compounds (VIP), the cumulative log_2_ Fold-Change (FC) values, and the most discriminant, significant (*p* < 0.05), and validated VIP compounds according to the ROC analysis.

Metabolic Pathways	n° of VIP Compounds	Cumulative log_2_FC	Most Discriminant VIP Compounds
MDP vs. MTX			
Tryptophan metabolism	8	1.52	4,6-Dihydroxyquinoline (VIP score: 2.458; AUC: 0.74; *p*-value: 4.7 × 10^−3^)
Sphingolipid metabolism	7	−1.31	L-Serine (VIP score: 2.372; AUC: 0.72; *p*-value: 8.8 × 10^−3^)
Glycerophospholipid metabolism	38	−17.46	PE(22:2(13Z,16Z)/16:0) (VIP score: 2.221; AUC: 0.69; *p*-value: 1.3 × 10^−2^)
MDP vs. CTR			
Glycerophospholipid metabolism	39	−24.47	PE(14:1(9Z)/18:4(6Z,9Z,12Z,15Z)) (VIP score: 2.205; AUC: 0.66; *p*-value: 2.8 × 10^−3^)
Pyrimidine metabolism	4	0.89	CDP (VIP score: 2.411; AUC: 0.77; *p*-value: 1.6 × 10^−3^)
MTX vs. CTR			
Sphingolipid metabolism	7	2.67	Ceramide (t18:0/16:0) (VIP score: 1.686; AUC: 0.69; *p*-value: 3.6 × 10^−2^)
Steroid hormones biosynthesis	12	0.27	Pregnanetriol (VIP score: 2.171; AUC: 0.67; *p*-value: 2.2 × 10^−2^)

## Data Availability

Appendix A is fully available at the reserved doi:https://doi.org/10.17632/vxn87nnkx4.1.

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
