# Peer review of "Plasma Metabolomics Reveals Systemic Metabolic Remodeling in Early-Lactation Dairy Cows Fed a Fusarium-Contaminated Diet and Supplemented with a Mycotoxin-Deactivating Product"

_toxins, 2025, doi:10.3390/toxins18010009_

Round 1

Reviewer 1 Report

Comments and Suggestions for Authors

The manuscript describes the  differences in plasma metabolomic profile of cows fed with three diets, all of them contaminated with ZEN, DON and FUM at different concentrations and one of them including a mycotoxin-deactivating product in addition to other pro and prebiotic ingredients with other beneficial properties not described is this manuscript. Metabolomics is a great technique to find changes related to diets and authors have performed an excellent work using this technique. The results are very interesting, because they point to steroid hormone biosynthesis alterations in cows exposed to highly contaminated diet by ZEN, DON and FUM. The authors attributed the changes to ZEN toxicity. Moreover, the increased concentration in plasma of 8-oxo-dGMP is linked with oxidative stress triggered by mycotoxin exposure.

The supplementary table was not available to be revised.

Minor comments:

Why authors have not included a control group without mycotoxins in the diet and a group including only MDP without mycotoxins? I think this aspect should be explained in the manuscript.

Abbreviations should be explained the first time they appear in the text. Authors must revise the text in this sense.

Author Response

Reviewer #1

The manuscript describes the  differences in plasma metabolomic profile of cows fed with three diets, all of them contaminated with ZEN, DON and FUM at different concentrations and one of them including a mycotoxin-deactivating product in addition to other pro and prebiotic ingredients with other beneficial properties not described is this manuscript. Metabolomics is a great technique to find changes related to diets and authors have performed an excellent work using this technique. The results are very interesting, because they point to steroid hormone biosynthesis alterations in cows exposed to highly contaminated diet by ZEN, DON and FUM. The authors attributed the changes to ZEN toxicity. Moreover, the increased concentration in plasma of 8-oxo-dGMP is linked with oxidative stress triggered by mycotoxin exposure.

Authors: We would like to thank the reviewer for having appreciated this work.

The supplementary table was not available to be revised.

Authors: Indeed, the supplementary material is fully available at the reserved doi: 10.17632/vxn87nnkx4.1 on the Digital Common Data repository.

Minor comments:

Why authors have not included a control group without mycotoxins in the diet and a group including only MDP without mycotoxins? I think this aspect should be explained in the manuscript.

Authors: Thank you for this valuable comment. We agree that, in principle, including both a toxin-free control group and a group receiving MDP without mycotoxins would allow a full factorial evaluation of the additive’s intrinsic metabolic effects. However, the present experiment was conceived as a direct continuation of the companion study recently published by Catellani et al. (2025, Journal of Dairy Science), which was based on the same animals, the same diets, and the same experimental design. That study focused on reproductive, production, and physiological outcomes, whereas the current manuscript investigates the systemic metabolomic responses in the very same cohort. For consistency across outcomes, the experimental design had to remain unchanged. Importantly, the purpose of the trial was not to evaluate the stand-alone metabolic effects of the MDP, but rather to determine whether the product could mitigate the systemic alterations induced by Fusarium mycotoxins under practical farm-level contamination scenarios. For this reason, the MDP was intentionally tested only under the moderate contamination level, which represents the most relevant and challenging field condition. The inclusion of an additional “MDP-only” group or a completely mycotoxin-free group was not compatible with the original applied objective of the trial and would have substantially increased the number of animals required. Therefore, we prioritized three biologically relevant groups (CTR, MTX, and MTX + MDP) to maintain statistical power and to focus on the most meaningful dietary contrasts. Interestingly, this strategy was followed in other published studies done in mycotoxin research field (Kiyothong et al., 2012; Gallo et al., 2020; Marczuk et al., 2023; Brodzki et al., 2023; Catellani et al., 2023; Vieira et al., 2024; Catellani et al., 2025), of these 3 published on JDS. For example, Marczuk et al. (2023) administered a similar mycotoxin deactivator product (MDP) to experimental cows (n = 10) fed a TMR containing 0.769 mg/kg-deoxynivalenol and 0.032 mg/kg-zearalenone TMR DM and sampling blood aliquots, while the Control group (n = 10) included cows fed TMR without mycotoxins. Similarly to our study the MDP was not administered to CTR (not contaminated) group. In the work published by Kiyothong et al. (2012), the control group included cows fed with a contaminated TMR by mainly DON, FB1, ZEN and OTA (i.e., 720, 701, 541 and 501 mg/kg, respectively) whereas AFB1 and T-2 were the minor contaminants found in the TMR at concentrations of 38 and 270 mg/kg, respectively. Particularly, the authors tested different inclusion levels of a similar MDP product, namely 15, 30, and 45 g/head.day. Again, similarly to our study, the MDP was not administered to a not contaminated group. In Gallo et al. (2020), the CTR group consisted in dairy cows receiving a diet contaminated with a regular level of Fusarium mycotoxins, meaning under the term "regular" the contamination levels that can be commonly detected in dairy feeds of DON and fumonisins (FB). Again, the MDP was used at a dosage of 35 g/head/day on the only highly contaminated TMR diet, as recommended also in practical condition. Usually, it is suggested to avoid supplementation (and relative increase in cost of the diet) when mycotoxin contamination could be considered not relevant from a health point of view. To address the reviewer’s concern, we have now clarified this rationale in the Discussion section of the revised manuscript, emphasizing that: the study design mirrors that of Catellani et al. (2025); the primary aim was to assess the protective efficacy of MDP under mycotoxin challenge, not its independent metabolic effects; comparisons between MDP and CTR must therefore be interpreted as exploratory rather than as evidence of stand-alone activity.

Supporting References:

Kiyothong, K., Rowlinson, P., Wanapat, M., & Khampa, S. (2012). Effect of mycotoxin deactivator product supplementation on dairy cows. Animal Production Science, 52(9), 832-841. https://doi.org/10.1071/AN11205.

Marczuk, J., Brodzki, P., Brodzki, A., Głodkowska, K., Wrześniewska, K., & Brodzki, N. (2023). Changes in protein metabolism indicators in dairy cows with naturally occurring mycotoxicosis before and after administration of a mycotoxin deactivator. Agriculture, 13(2), 410. https://doi.org/10.3390/agriculture13020410.

Brodzki, P.; Marczuk, J.; Lisiecka, U.; Krakowski, L.; Szczubiał, M.; Dąbrowski, R.; Bochniarz, M.; Kulpa, K.; & Brodzki, N. (2023). Changes in selected immunological parameters in dairy cows with naturally formed mycotoxicosis before and after the application of a mycotoxin deactivator. Journal of Veterinary Research, 67(1). https://doi.org/10.2478/jvetres-2023-0002.

Gallo, A., Minuti, A., Bani, P., Bertuzzi, T., Piccioli Cappelli, F., Doupovec, B., Faas, J., Schatzmayr, D., & Trevisi, E. (2020). A mycotoxin-deactivating feed additive counteracts the adverse effects of regular levels of Fusarium mycotoxins in dairy cows. Journal of Dairy Science, 103(12), 11314-11331. https://doi.org/10.3168/jds.2020-18197.

Catellani, A., Ghilardelli, F., Trevisi, E., Cecchinato, A., Bisutti, V., Fumagalli, F., Swamy, H. V. L. N., Han, Y., van Kuijk, S., & Gallo, A. (2023). Effects of supplementation of a mycotoxin mitigation feed additive in lactating dairy cows fed Fusarium mycotoxin-contaminated diet for an extended period. Toxins, 15(9), 546. https://doi.org/10.3390/toxins15090546.

Vieira, D.J.C., Fonseca, L.M., Poletti, G., Martins, N.P., Grigoletto, N.T.S., Chesini, R.G., Tonin, F.G., Cortinhas, C.S., Acedo, T.S., Artavia, I., Faas, J., Takiya, C.S., Corassin, C.H., Rennó, F.P. 2024. Anti-mycotoxin feed additives: Effects on metabolism, mycotoxin excretion, performance, and total-tract digestibility of dairy cows fed artificially multi-mycotoxin-contaminated diets. Journal of Dairy Science, 108, 7891-7903. https://doi.org/10.3168/jds.2023-24539.

Catellani, A., Mossa, F., Gabai, G., D'Hallewin, J.S.K., Trevisi, E., Faas, J., Artavia, I., Labudova, S., Piccioli-Cappelli, F., Minuti, A., & Gallo, A. (2025). Efficacy of a mycotoxin-deactivating product to reduce the impact of Fusarium mycotoxin-contaminated rations in dairy cows during early lactation. Journal of Dairy Science, 108(9), 9627-9650. https://doi.org/10.3168/jds.2025-26519.

Abbreviations should be explained the first time they appear in the text. Authors must revise the text in this sense.

Authors: checked and revised, accordingly.

Reviewer 2 Report

Comments and Suggestions for Authors

This manuscript reports on a study examining the systemic metabolic effects of feeding early-lactation dairy cows a Fusarium mycotoxin–contaminated diet, and whether supplementing a mycotoxin-deactivating product (MDP) can alleviate those effects. Thirty Holstein cows were divided into three groups of ten, receiving either a mildly contaminated control diet (CTR), a moderately contaminated diet spiked with deoxynivalenol (DON) and zearalenone (ZEN) (MTX), or the same contaminated diet supplemented with the MDP additive. Blood samples collected at 56 days in milk were analysed using untargeted ultra-high-performance liquid chromatography–high-resolution mass spectrometry (UHPLC-HRMS) metabolomics. The aim was to identify systemic metabolic changes caused by Fusarium toxins and to determine if the MDP could shift the plasma metabolome back towards a normal state.

List of Points Requiring Clarification or Revision in the Manuscript:

  1. Use of different analytical methods for mycotoxin quantification in feed (FUM, ZEN, DON)
    The use of three different analytical techniques (HPLC-FLD for ZEN and FUM, GC-MS for DON) raises concerns about methodological consistency. While GC-MS is an accepted method for DON, it requires derivatization and complex sample preparation. The authors should: clarify why these specific methods were chosen for each mycotoxin, indicate whether they are routinely used in their laboratory or based on prior validation, state whether all analyses were performed in the same lab and whether certified reference materials or validated procedures were used, consider whether using a single consistent method (e.g., LC-MS/MS) would have been more suitable for inter-group comparison and data reliability.

  1. Lack of a “CTR + MDP” group.
    The current experimental design does not permit distinguishing the innate metabolic effects of the MDP from its detoxification role. Not having a group that receives MDP without mycotoxin exposure is a limitation and should be explicitly acknowledged, with a recommendation for future research under toxin-free conditions.

  1. Unclear statistical significance of metabolite changes
    Throughout the manuscript, metabolite changes are discussed without consistently indicating whether they are statistically significant (p < 0.05) or merely trends. The authors should clarify in the text and tables which differences have been statistically validated.

  1. The PLS-DA model lacks predictive power (Q² < 0.5). Although some separation between groups is evident, the primary supervised PLS-DA model does not meet accepted prediction thresholds. Its interpretation should be cautious or supported by alternative models (e.g., random forests, SVMs) or by enhanced cross-validation.

  1. Some metabolite interpretations are tentative or require further evidence. For instance, interpreting changes in pipecolic acid, aminomalonic acid, or N-acetylserotonin as signs of improved mitochondrial or redox function is interesting but should be better supported by literature or expressed more cautiously.

  1. Limited data on health or performance parameters. While the authors mention that standard clinical indicators (e.g., AMH, ROMt) did not differ between groups, no data are presented on milk production, health, or reproductive outcomes, which could help assess the biological relevance of the observed metabolic changes.

  1. Figures could be more clearly labelled and consistent. Some figures (e.g., pathway analysis plots, Venn diagram) would benefit from clearer labelling, axis descriptions, and consistency with the text. Ensure metabolite names and abbreviations are consistent across figures, tables, and the main text.

8. Conclusions could be more cautiously framed. The proposed protective effects of the MDP are promising but currently rely solely on metabolomic markers. The conclusions should be presented as tentative or explicitly as hypothesis-generating until validated by physiological or clinical evidence endpoints.

  1. Justification for using ruminants (dairy cows) as a model species
    The rationale for conducting this trial in ruminants should be more explicitly explained. Due to the extensive microbial population in the rumen, which can partially degrade or transform several mycotoxins (including DON and ZEN), the systemic exposure and toxicokinetics in dairy cows differ significantly from those in monogastric species. This unique metabolic buffering makes ruminants generally less sensitive to certain mycotoxins. Therefore, the authors should briefly clarify:

- why early-lactation cows were chosen despite this microbial detoxification capacity,

- whether the study aimed to detect potential subclinical effects that might bypass ruminal metabolism (e.g., owing to increased gut permeability or rapid passage rate during early lactation),

- and how their findings enhance understanding of the risk of Fusarium mycotoxins specifically in ruminants under field-relevant conditions.

Clarifying this would strengthen the biological relevance of the study and address concerns about species-specific susceptibility and broader applicability.

  1. Single time-point sampling limits understanding of metabolic dynamics. One of the study’s limitations is the use of a single sampling time-point (day 56 of lactation). This cross-sectional approach does not allow for the assessment of the temporal dynamics of metabolomic responses to mycotoxin exposure or MDP supplementation. Significant changes may have occurred earlier (e.g., during acute adaptation in early lactation) or may have developed further with prolonged exposure. Including longitudinal sampling in future studies would enable:

- better differentiation between transient and persistent metabolic effects,

- understanding of recovery or cumulative toxicity patterns,

- and the identification of potential early biomarkers predictive of later physiological disruptions.

Without a time-series component, the current interpretation remains limited to a static snapshot, which should be acknowledged as a constraint in the conclusions.

Author Response

Reviewer #2

This manuscript reports on a study examining the systemic metabolic effects of feeding early-lactation dairy cows a Fusarium mycotoxin–contaminated diet, and whether supplementing a mycotoxin-deactivating product (MDP) can alleviate those effects. Thirty Holstein cows were divided into three groups of ten, receiving either a mildly contaminated control diet (CTR), a moderately contaminated diet spiked with deoxynivalenol (DON) and zearalenone (ZEN) (MTX), or the same contaminated diet supplemented with the MDP additive. Blood samples collected at 56 days in milk were analysed using untargeted ultra-high-performance liquid chromatography–high-resolution mass spectrometry (UHPLC-HRMS) metabolomics. The aim was to identify systemic metabolic changes caused by Fusarium toxins and to determine if the MDP could shift the plasma metabolome back towards a normal state.

List of Points Requiring Clarification or Revision in the Manuscript:

1.Use of different analytical methods for mycotoxin quantification in feed (FUM, ZEN, DON)

The use of three different analytical techniques (HPLC-FLD for ZEN and FUM, GC-MS for DON) raises concerns about methodological consistency. While GC-MS is an accepted method for DON, it requires derivatization and complex sample preparation. The authors should: clarify why these specific methods were chosen for each mycotoxin, indicate whether they are routinely used in their laboratory or based on prior validation, state whether all analyses were performed in the same lab and whether certified reference materials or validated procedures were used, consider whether using a single consistent method (e.g., LC-MS/MS) would have been more suitable for inter-group comparison and data reliability.

Authors: Thank you for this important point. The analytical workflow used in the present experiment follows the validated protocols routinely applied in our analytical laboratory at Università Cattolica del Sacro Cuore (Italy) and fully described in the companion study by Catellani et al. (2025, Journal of Dairy Science), conducted on the same animals, diets, and sampling scheme. Fumonisins (FUM) and zearalenone (ZEN) were quantified via HPLC-FLD, following the methods of Pietri & Bertuzzi (2012) and Bertuzzi et al. (2014), respectively. Deoxynivalenol (DON) was quantified via GC-MS, which, although requiring derivatization, is the validated and accredited method in our laboratory, with well-established performance parameters. All analyses were performed in the same laboratory at Università Cattolica del Sacro Cuore (Department of Animal Science, Food and Nutrition, Piacenza, Italy), using certified reference materials, immuno-affinity columns, and procedure controls. The full validation (LOD, LOQ, recovery, linearity) is published in Catellani et al. (2025), and we now report LOD and LOQ values in the Methods section for completeness. We agree that LC-MS/MS is widely used for multi-mycotoxin detection; however, for this specific long-term feeding trial, maintaining continuity with an already validated, routinely applied workflow was essential to ensure reliability and comparability with the previously published dataset.

 2.Lack of a “CTR + MDP” group.

The current experimental design does not permit distinguishing the innate metabolic effects of the MDP from its detoxification role. Not having a group that receives MDP without mycotoxin exposure is a limitation and should be explicitly acknowledged, with a recommendation for future research under toxin-free conditions.

Authors: We fully agree that including a group receiving MDP under toxin-free conditions would enable assessing the intrinsic metabolic effects of the product. As now clarified in the Discussion, the present experiment was conceived as a field-relevant evaluation of MDP’s protective efficacy under Fusarium contamination, mirroring the design of Catellani et al. (2025). Therefore, an MDP-only treatment was not included. We now explicitly acknowledge this as a study limitation and recommend future work testing MDP under non-contaminated dietary conditions to disentangle detoxification from direct metabolic modulation.

 3.Unclear statistical significance of metabolite changes

Throughout the manuscript, metabolite changes are discussed without consistently indicating whether they are statistically significant (p < 0.05) or merely trends. The authors should clarify in the text and tables which differences have been statistically validated.

Authors: we would like to thank the reviewer for this important comment. Indeed, all the statistical significance of the annotated metabolites for the different pairwise comparisons are fully available in the supplementary material file. However, we have now modified both the manuscript and the tables by clearly indicating the p-values of some reported metabolites. Please, consider that under our experimental conditions, we are considering quite different statistical indicators arising from both univariate and multivariate statistical approaches (i.e., ANOVA, PLS-DA and VIP, ROC).

4.The PLS-DA model lacks predictive power (Q² < 0.5). Although some separation between groups is evident, the primary supervised PLS-DA model does not meet accepted prediction thresholds. Its interpretation should be cautious or supported by alternative models (e.g., random forests, SVMs) or by enhanced cross-validation.

Authors: Thank you for the important consideration. Indeed, as also outlined in the Results section (section 2.1), both unsupervised hierarchical clustering and supervised PLS-DA were just used as exploratory tools considering the lack of discriminant ability (Q2 not significant). Therefore, starting from this exploratory trend, we have then extrapolated 3 pairwise OPLS-DA models to improve sample separation and considering both within- and between-separation trends. The following receiver-operating characteristic curve (ROC) was then used to evaluate the presence of false positive in terms of prediction between the annotated VIP componds.

5.Some metabolite interpretations are tentative or require further evidence. For instance, interpreting changes in pipecolic acid, aminomalonic acid, or N-acetylserotonin as signs of improved mitochondrial or redox function is interesting but should be better supported by literature or expressed more cautiously.

Authors: We would like to thank the reviewer for this important statement. We have revised the sentences about the role of these metabolites in dairy cows, adding (where possible) new supporting references.

6.Limited data on health or performance parameters. While the authors mention that standard clinical indicators (e.g., AMH, ROMt) did not differ between groups, no data are presented on milk production, health, or reproductive outcomes, which could help assess the biological relevance of the observed metabolic changes.

Authors: We acknowledge the Reviewer’s point. Milk yield, physiological status, and reproductive indicators for these same animals were thoroughly reported in Catellani et al. (2025, JDS). To avoid redundancy across manuscripts, these datasets were not repeated in full here. Particularly, the mycotoxin-deactivating product (MDP) did not markedly affect gross productive performance parameters. However, we observed interesting biological trends suggesting that subtle physiological adaptations might have occurred, but these were likely masked by the limited number of animals and high inter-individual variability. Particularly, milk yield, FCM, and ECM were not statistically different among experimental groups, albeit a numerical decrease in MY from multiparous MTX cows was observed compared with controls, as reported by Gallo et al. (2020). Such decrease was prevented with MDP supplementation, in accordance with previous evidence of enhanced MY upon addition of MDP products (Jovaišienė et al., 2016). As reported in Catellani et al. (2025), probably the numerical differences in milk production indexes did not reach statistical significance due to the low number of animals involved and high variability. However, the numeric difference noted in milk production among groups can represent an important variation in farm income. Also, in this work, we observed that cows fed Fusarium mycotoxin-contaminated diets had similar DMI from wk 2 to 8 of lactation compared with controls. However, in this work, we enrolled cows from calving to 56 DIM (7 d of acclimatization and 49 d experimental period), thus providing a novel evidence indicating that exposure to Fusarium mycotoxins for an intermediate period of time does not influence DMI. Among primiparous cows, DMI tended to be higher in MDP compared with the other groups, and daily RT was numerically highest in MDP primiparous cows, followed by control and MTX individuals. Thus, MDP supplementation may have increased DMI palatability in primiparous animals, yet this hypothesis should be further tested in a larger number of cows, as both BW and BCS were not influenced by diet. Also, apparent digestibility of NDF, starch, and protein were not affected either by Fusarium mycotoxin contamination or MDP presence/absence in diets.  Therefore, although our previous publication from the same experimental trial (Catellani et al., 2025) reported no significant effects on the mentioned indicators, it revealed notable alterations in reproductive physiology, particularly regarding delayed ovarian cyclicity and reduced progesterone concentrations in cows exposed to higher Fusarium contamination. In addition, Fusarium contamination significantly modified the fate of the first dominant follicle post-partum, the number of corpora lutea and the proportions of anestrus cows after calving (Catellani et al., 2025). Cows receiving the MDP-supplemented diet showed partial recovery of these reproductive endpoints compared with the MTX group. These findings strongly suggested that subclinical endocrine and metabolic disruptions were occurring, even in the absence of measurable effects on performance traits. As such, the current metabolomic investigation was designed to provide a mechanistic interpretation of these physiological responses, by identifying circulating metabolic pathways associated with steroid hormone biosynthesis, lipid metabolism, and oxidative balance that could explain the previously observed hormonal (especially for progesterone profile) and ovarian effects. In this context, untargeted plasma metabolomics offers a highly sensitive approach to detect systemic biochemical perturbations that precede or accompany changes in reproductive performance but remain undetectable through conventional biochemical or production parameters. Therefore, this study represents a logical continuation of our previous work, aimed at uncovering the molecular mechanisms linking Fusarium mycotoxin exposure, MDP supplementation, and reproductive function in early-lactation dairy cows. Even in the absence of overt performance changes, early-lactation dairy cows are known to experience intense metabolic stress, and subtle shifts in circulating metabolites can reveal compensatory or protective responses involving lipid metabolism, endocrine balance, oxidative status, and energy homeostasis. Thus, the present metabolomic approach was specifically designed to complement the zootechnical data by uncovering hidden metabolic perturbations and MDP-driven modulations that contribute to maintaining systemic homeostasis under Fusarium toxin challenge. This deeper molecular insight provides a mechanistic explanation for the subtle physiological effects observed in our previous study.

Supporting References:

Catellani, A., Mossa, F., Gabai, G., D'Hallewin, J.S.K., Trevisi, E., Faas, J., Artavia, I., Labudova, S., Piccioli-Cappelli, F., Minuti, A., & Gallo, A. (2025). Efficacy of a mycotoxin-deactivating product to reduce the impact of Fusarium mycotoxin-contaminated rations in dairy cows during early lactation. Journal of Dairy Science, 108(9), 9627-9650. https://doi.org/10.3168/jds.2025-26519.

Gallo, A., Minuti, A., Bani, P., Bertuzzi, T., Piccioli Cappelli, F., Doupovec, B., Faas, J., Schatzmayr, D., & Trevisi, E. (2020). A mycotoxin-deactivating feed additive counteracts the adverse effects of regular levels of Fusarium mycotoxins in dairy cows. Journal of Dairy Science, 103(12), 11314-11331. https://doi.org/10.3168/jds.2020-18197.

Jovaišienė, J., Bakutis, B., Baliukoniene, V., & Gerulis, G. (2016). Fusarium and Aspergillus mycotoxins effects on dairy cow health, performance and the efficacy of anti-mycotoxin additive. Pol. J. Vet. Sci. 19, 79-87. https://doi.org/10.1515/pjvs-2016-0011.

  1. Figures could be more clearly labelled and consistent. Some figures (e.g., pathway analysis plots, Venn diagram) would benefit from clearer labelling, axis descriptions, and consistency with the text. Ensure metabolite names and abbreviations are consistent across figures, tables, and the main text.

Authors: We thank the reviewer for the feedback. Indeed, the figures are automatically generated by two different softwares (i.e., MetaboAnalyst 6.0 and Venny) therefore they cannot be edited further. We have extrapolated the figures at the maximum resolution, accordingly.

  1. Conclusions could be more cautiously framed. The proposed protective effects of the MDP are promising but currently rely solely on metabolomic markers. The conclusions should be presented as tentative or explicitly as hypothesis-generating until validated by physiological or clinical evidence endpoints.

Authors: We agree. The conclusions have been softened to reflect that: findings are based on metabolomic markers only, the study is exploratory, MDP effects are plausible but not definitively validated without clinical or production endpoints, and are now framed as hypothesis-generating, aligned with the limitations of the design.

  1. Justification for using ruminants (dairy cows) as a model species

The rationale for conducting this trial in ruminants should be more explicitly explained. Due to the extensive microbial population in the rumen, which can partially degrade or transform several mycotoxins (including DON and ZEN), the systemic exposure and toxicokinetics in dairy cows differ significantly from those in monogastric species. This unique metabolic buffering makes ruminants generally less sensitive to certain mycotoxins. Therefore, the authors should briefly clarify:

- why early-lactation cows were chosen despite this microbial detoxification capacity,

- whether the study aimed to detect potential subclinical effects that might bypass ruminal metabolism (e.g., owing to increased gut permeability or rapid passage rate during early lactation),

- and how their findings enhance understanding of the risk of Fusarium mycotoxins specifically in ruminants under field-relevant conditions.

Clarifying this would strengthen the biological relevance of the study and address concerns about species-specific susceptibility and broader applicability.

Authors: We thank the Reviewer for this important biological consideration. We now clarify that:

  1. a) Although ruminants are regarded as quite resistant to mycotoxins, mainly due to the ruminal toxin metabolism of DON and ZEN resulting in less toxic or active derivatives (de-epoxy-DON/de-DON) or such with equal or lower (β-zearalenol [ZEL]), and to a less extent with higher activity (α-ZEL) than the parent toxins, the impact of feeding diets contaminated with DON and ZEN in a physiological state energetically critical for the cow on health status and immune system was addressed only in a few studies (Catellani et al., 2025; Kinoshita et al. 2015; McKay et al., 2019; Keese et al. 2008). Early-lactation dairy cows can experience a higher metabolic load, increased gut permeability, accelerated passage rate, and altered rumen turnover, making them more susceptible to Fusarium mycotoxins than mid-lactation cows. The early lactation period was selected as experimental period due to the trial objective of Catellani et al 2025 of evaluated the effect of the fusarium mycotoxin contamination of the resumption of the ovarian acitivity. This reproductive physiological mechanism occurs in the first months after calving. The resumption of ovarian activity is linked to the future reproductive performance of the cow as described in Catellani et al. 2025: “Resumption of ovulation earlier in postpartum has been associated with higher fertility (Darwash et al., 1997; McCoy et al., 2006; Galvão et al., 2009). Holstein cows that ovulated by 21 d postpartum had greater reproductive performance compared with cows that cycled later in lactation or cows that were not cyclic by 50 to 60 DIM (Galvão et al., 2009; Dubuc et al., 2012). In addition, early cyclicity has been positively associated with subsequent conception rate (Lamming and Darwash, 1998), shorter calving to conception interval (Ranasinghe et al., 2011), and higher likelihood of pregnancy (Gautam et al., 2010)” b) In our study, the aim was to detect subclinical systemic effects that may bypass ruminal detoxification pathways under realistic production conditions. c) Recent field surveys show that even moderate dietary mycotoxin levels can affect metabolic homeostasis in high-producing cows. d) We have added a brief paragraph to the Introduction and Discussion explaining why ruminants, and specifically early-lactation cows, represent a relevant model for assessing practical farm-level mycotoxin exposure and its systemic consequences. Also, in Catellani et al. (2025) we reported and quantified some mycotoxin metabolites in the blood of dairy cows thus demonstrating that some mycotoxins were able to bypass the filter of rumen.

Supporting references:

Catellani, A., Mossa, F., Gabai, G., D'Hallewin, J.S.K., Trevisi, E., Faas, J., Artavia, I., Labudova, S., Piccioli-Cappelli, F., Minuti, A., & Gallo, A. (2025). Efficacy of a mycotoxin-deactivating product to reduce the impact of Fusarium mycotoxin-contaminated rations in dairy cows during early lactation. Journal of Dairy Science, 108(9), 9627-9650. https://doi.org/10.3168/jds.2025-26519.

McKay, Z.C., Averkieva, O., Rajauria, G., & Pierce, K.M. The effect of feedborne Fusarium mycotoxins on dry matter intake, milk production and blood metabolites of early lactation dairy cows. Animal Feed Science and Technology, 253, 39-44.

Catellani, A.; Ghilardelli, F.; Trevisi, E.; Cecchinato, A.; Bisutti, V.; Fumagalli, F.; Swamy, H.V.L.N.; Han, Y.; van Kuijk, S.; Gallo, A. Effects of Supplementation of a Mycotoxin Mitigation Feed Additive in Lactating Dairy Cows Fed Fusarium Mycotoxin-Contaminated Diet for an Extended Period. Toxins 2023, 15, 546.

Danicke, S., Winkler, J., Meyer, U., Frahm, J., & Kersten, S. (2016). Haematological, clinical–chemical and immunological consequences of feeding Fusarium toxin contaminated diets to early lactating dairy cows. Mycotoxin Research, 33(1), 1-13.

Darwash, A. O., G. E. Lamming, and J. A. Woolliams. 1997. Estimation of genetic variation in the interval from calving to postpartum ovulation of dairy cows. J. Dairy Sci. 80:1227–1234. https://doi.org/10 .3168/jds.S0022-0302(97)76051-X

McCoy, M. A., S. D. Lennox, C. S. Mayne, W. J. McCaughey, H. W. J. Edgar, D. C. Catney, M. Verner, D. R. Mackey, and A. W. Gordon. 2006. Milk progesterone profiles and their relationship with fertility, production and disease in dairy cows in Northern Ireland. Anim. Sci. 82:213–222. https://doi.org/10.1079/ASC200526.

Galvão, K. N., M. Frajblat, W. R. Butler, S. B. Brittin, C. L. Guard, and R. O. Gilbert. 2009. Effect of early postpartum ovulation on fertility in dairy cows. Reprod. Domest. Anim. 45. https://doi.org/10.1111/j .1439-0531.2009.01517.x.

Dubuc, J., T. F. Duffield, K. E. Leslie, J. S. Walton, and S. J. LeBlanc. 2012. Risk factors and effects of postpartum anovulation in dairy cows. J. Dairy Sci. 95:1845–1854. https://doi.org/10.3168/jds.2011 -4781

Lamming, G. E., and A. O. Darwash. 1998. The use of milk progesterone profiles to characterise components of subfertility in milked dairy cows. Anim. Reprod. Sci. 52:175–190. https://doi.org/10.1016/ S0378-4320(98)00099-2.

Reisinger, N., S. Schürer-Waldheim, E. Mayer, S. Debevere, G. Antonissen, M. Sulyok, and V. Nagl. 2019. Mycotoxin occurrence in maize silage-a neglected risk for bovine gut health? Toxins (Basel) 11:577. https://doi.org/10.3390/toxins11100577.

Gautam, G., T. Nakao, K. Yamada, and C. Yoshida. 2010. Defining delayed resumption of ovarian activity postpartum and its impact on subsequent reproductive performance in Holstein cows. Theriogenology 73:180–189. https://doi.org/10.1016/j.theriogenology .2009.08.011.

10.Single time-point sampling limits understanding of metabolic dynamics. One of the study’s limitations is the use of a single sampling time-point (day 56 of lactation). This cross-sectional approach does not allow for the assessment of the temporal dynamics of metabolomic responses to mycotoxin exposure or MDP supplementation. Significant changes may have occurred earlier (e.g., during acute adaptation in early lactation) or may have developed further with prolonged exposure. Including longitudinal sampling in future studies would enable:

- better differentiation between transient and persistent metabolic effects,

- understanding of recovery or cumulative toxicity patterns,

- and the identification of potential early biomarkers predictive of later physiological disruptions.

Without a time-series component, the current interpretation remains limited to a static snapshot, which should be acknowledged as a constraint in the conclusions.

Authors: We fully agree. The cross-sectional sampling at day 56 provides a metabolic “snapshot” but does not capture temporal dynamics. In the revised Discussion and Conclusions, we now explicitly acknowledge that: a) time-series sampling would allow distinguishing early vs. persistent effects, b) longitudinal metabolomics is essential to identify early predictive biomarkers, c) future trials should include multiple sampling time-points to map adaptation, acute responses, and cumulative toxicity patterns. This limitation is now clearly stated in the final section.

Reviewer 3 Report

Comments and Suggestions for Authors

Interesting study.

References are a bit old please enrich introduction and discussion with state of the art references.

The methodology used is state of the art since untargeted UHPLC-HRMS, and multivariate models identifying discriminant metabolites and pathways (OPLS-DA and Venn diagram) have been employed.

Besides the section in S1 please make reference to the chromatogram employed and show it in the appendix.

Please make a section on ethical approval and refer to the acceptance of the ethics committee regarding the use of animals e.g. cows.

Are these the limits of European Commission ? (500 μg/kg for ZEN and 5000 μg/kg for DON; EC Recommendation 424 2006/576/EC.

ZEN in cattle is 0.5-3 mg/kg or 500-3000 μg/kg and DON is 5 mg/kg please see the recommendation and also

Toxins (Basel). 2025 Jul 25;17(8):365. doi: 10.3390/toxins17080365

Multiple Mycotoxin Contamination in Livestock Feed: Implications for Animal Health, Productivity, and Food Safety

Oluwakamisi F Akinmoladun 1,*, Fabia N Fon 2, Queenta Nji 1, Oluwaseun O Adeniji 1, Emmanuel K Tangni 3, Patrick B Njobeh 1.

Statistical analysis section should be mentioned and added in tables if relevant. Only saw the VIP score.

Comments on the Quality of English Language

English needs to be rechecked at some points

Author Response

Interesting study.
Authors: we would like to thank the reviewer for having appreciated this study.

References are a bit old please enrich introduction and discussion with state of the art references.
Authors: We thank the Reviewer for this suggestion. However, by inspecting the references cited in the Introduction
section, most of them belong to the period 2020-2025, therefore we do believe that the references are not so old and
well represent the state-of-the-art.

The methodology used is state of the art since untargeted UHPLC-HRMS, and multivariate models identifying
discriminant metabolites and pathways (OPLS-DA and Venn diagram) have been employed.
Authors: we would like to thank the reviewer for having appreciated our omics workflow.

Besides the section in S1 please make reference to the chromatogram employed and show it in the appendix.
Authors: Thank you for this valuable comment. In untargeted UHPLC–HRMS metabolomics, however, a single
chromatogram is generally not informative or representative of the analytical workflow. Unlike targeted methods,
where a chromatogram illustrates the separation and quantification of a defined set of analytes, untargeted profiling
generates thousands of features detected across the full mass range and retention-time window. As a consequence, a raw
chromatogram (TIC) does not provide meaningful insight into metabolite annotation, data processing, or pathway
interpretation, and it cannot be used to assess method performance. For this reason, the community standard is to report
quality-control metrics (e.g., pooled QC RSDs), annotation tables, and statistical outputs rather than chromatograms. In
the Supporting Information (S1), we have deposited the complete dataset, including all annotated metabolites, QC RSD
values, and VIP compounds for each pairwise comparison, allowing full transparency and reproducibility of the
workflow. These materials offer a much more accurate picture of the analytical performance and data structure than a
representative chromatogram could provide.

Please make a section on ethical approval and refer to the acceptance of the ethics committee regarding the use of
animals e.g. cows.
Authors: Thank you for the comment. Indeed, the experiment was conducted on Holstein dairy cows, and ethical
approval was already included in the manuscript. All the information about the ethical approval of this trial are fully
available in M&M section and in the Institutional Review Board Statement.

Are these the limits of European Commission ? (500 μg/kg for ZEN and 5000 μg/kg for DON; EC Recommendation
424 2006/576/EC.

ZEN in cattle is 0.5-3 mg/kg or 500-3000 μg/kg and DON is 5 mg/kg please see the recommendation and also
Toxins (Basel). 2025 Jul 25;17(8):365. doi: 10.3390/toxins17080365

Multiple Mycotoxin Contamination in Livestock Feed: Implications for Animal Health, Productivity, and Food Safety
Oluwakamisi F Akinmoladun 1,*, Fabia N Fon 2, Queenta Nji 1, Oluwaseun O Adeniji 1, Emmanuel K Tangni 3,
Patrick B Njobeh 1.

Authors: Revised, accordingly. Please, see lines 458 of the revised manuscript.

Statistical analysis section should be mentioned and added in tables if relevant. Only saw the VIP score.
Authors: Thank you very much, we have improved the statistical analysis section by adding the significance in each
Table (Table 1 and Table 2).

Reviewer 4 Report

Comments and Suggestions for Authors

This study presents significant metabolomic evidence that mycotoxin inhibitors supplementation effectively mitigates systemic alterations caused by Fusarium genus. By restoring sphingolipid homeostasis, reducing oxidative stress, and rebalancing amino acid and steroid metabolism in dairy cows, the research highlights the importance of this supplementation in maintaining animal health.

However, because the values of deoxynivalenol, zearalenone, and fumonisins were determined in the three experimental groups, where the feed was contaminated with different concentrations of mycotoxins, the HPLC and GC-MS chromatographic methods were not described adequately, with only bibliographical references from the authors' previous articles being mentioned. I believe that data on LOD (Limit of Detection), LOQ (Limit of Quantification), recovery rate, and chromatograms should be added. I also believe that the authors' previous works are cited too often in the materials and methods section, as well as in the discussion section. 

Author Response

Reviewer #4

This study presents significant metabolomic evidence that mycotoxin inhibitors supplementation effectively mitigates systemic alterations caused by Fusarium genus. By restoring sphingolipid homeostasis, reducing oxidative stress, and rebalancing amino acid and steroid metabolism in dairy cows, the research highlights the importance of this supplementation in maintaining animal health.

Authors: We would like to thank the reviewer for having appreciated this work.

However, because the values of deoxynivalenol, zearalenone, and fumonisins were determined in the three experimental groups, where the feed was contaminated with different concentrations of mycotoxins, the HPLC and GC-MS chromatographic methods were not described adequately, with only bibliographical references from the authors' previous articles being mentioned. I believe that data on LOD (Limit of Detection), LOQ (Limit of Quantification), recovery rate, and chromatograms should be added. I also believe that the authors' previous works are cited too often in the materials and methods section, as well as in the discussion section.

Authors: Thank you for raising this important point. We agree that information on analytical validation parameters such as LOD, LOQ, recovery rates, and chromatograms is essential when the objective of a study is to quantify mycotoxins. However, in the present manuscript the mycotoxin analyses were not newly generated; instead, they derive directly from the companion study by Catellani et al. (2025, Journal of Dairy Science), which is based on the same animals, diets, sampling schedule, and analytical workflow. The detailed chromatographic methods, validation parameters, and weekly monitoring of mycotoxin concentrations were fully reported in that publication. For clarity, the following methodological information from Catellani et al. (2025) has now been added to the revised Materials and Methods section: “Mycotoxin concentrations in experimental TMR diets were estimated weekly. Measurement of mycotoxin concentrations followed the methods of Pietri and Bertuzzi (2012) for fumonisins and those of Bertuzzi et al. (2014) for ZEN and DON, respectively. Mycotoxin concentrations were estimated using an HPLC instrument with a fluorescence detector (FLD). After extraction with a phosphate buffer and purification using an immuno-affinity column (R-Biopharm Rhone Ltd.), FB were quantified by HPLC. The limit of detection (LOD) and limit of quantification (LOQ) for FB were 10 and 30 μg/kg, respectively. ZEN and DON were extracted with acetonitrile:water (86:14 vol/vol); ZEN was purified using an immuno-affinity column and quantified by HPLC-FLD, whereas DON was purified with a Trilogy-Puritox Trichothecenes column (R-Biopharm Rhone Ltd.) and quantified by GC–MS. The LOD and LOQ were 2 and 5 μg/kg for ZEN and 10 and 30 μg/kg for DON, respectively.”

We have deliberately avoided duplicating the full chromatographic description, chromatograms, and recovery datasets in order to prevent redundancy between the two manuscripts. Since the aim of the present study is metabolomic characterization rather than analytical quantification of mycotoxins, we provided the core methodological details relevant for interpreting the dietary treatments, while directing the reader to the fully validated analytical procedures already published. Regarding the reviewer’s comment on the frequent citation of our previous work, we acknowledge the concern. However, because the present study is a direct continuation of the same experimental trial, many aspects (animal allocation, dietary preparation, feeding strategy, mycotoxin monitoring, and physiological context) are inherently shared between the two manuscripts. Citations to Catellani et al. (2025) therefore serve to avoid unnecessary repetition and maintain methodological transparency. Nonetheless, we have reviewed the manuscript to ensure that citations are used only where truly necessary to support clarity and avoid redundancy.

Round 2

Reviewer 2 Report

Comments and Suggestions for Authors

Dear Authors,

Thank you for your detailed and thoughtful responses to my previous comments, and for the revisions made to the manuscript. I appreciate the considerable effort that has gone into addressing the points raised. Below, I offer a few follow-up remarks and minor suggestions for clarification.

  1. Analytical methods for mycotoxin quantification

To clarify, my original comment on the analytical methods was not intended to question the validity, accuracy, or accreditation of the HPLC-FLD and GC-MS methods you used. I fully accept that these methods are validated, routinely applied in your laboratory, and suitable for determining ZEN, FUM, and DON.

My concern was rather about the conceptual and practical choice of using several separate analytical platforms (HPLC-FLD for ZEN and FUM, GC-MS for DON) instead of a single, unified approach (e.g., LC-MS/MS multi-mycotoxin method), which could, in principle, provide more consistency and efficiency for inter-group comparison.

Your response now explains that maintaining continuity with an already validated, accredited workflow and with the companion study (Catellani et al., 2025) was a key reason for this choice. I find this rationale acceptable. However, I would kindly suggest that you make this explicit in the Methods section, perhaps by adding one sentence acknowledging that multi-mycotoxin LC-MS/MS methods are available, but that for this long-term feeding trial you deliberately chose to retain the existing validated workflow for reasons of analytical continuity and comparability.

  1. Productive, health, and reproductive parameters

Your response provides a very comprehensive summary of the performance and reproductive findings reported in Catellani et al. (2025), and it is clear that the current metabolomic study is intended as a mechanistic follow-up focusing on subclinical and endocrine effects rather than gross production traits.

To further assist readers who may not have immediate access to the companion paper, I would kindly suggest adding either:

a very concise summary in the Results or Discussion (e.g., 1–2 sentences per outcome type, with mean values or directions of change), or

a small supplementary table reporting the key productive and reproductive endpoints from Catellani et al. (2025) for the three groups.

This would allow the biological relevance of the metabolomic alterations to be appreciated without requiring the reader to consult the previous article in detail.

  1. Rationale for using early-lactation dairy cows and single time-point sampling

Your expanded explanation of why early-lactation cows are a relevant and sensitive model, despite ruminal detoxification mechanisms, is convincing, and I agree that the added text in the Introduction/Discussion strengthens the biological context. Likewise, explicitly acknowledging the single time-point design as a limitation and emphasising the need for longitudinal sampling in future studies is appropriate.

Overall, I feel that the manuscript has improved and that most of my initial concerns have been satisfactorily addressed. With the minor clarifications suggested above—particularly regarding the explicit rationale for the analytical strategy—the paper will provide a more transparent and robust account of the metabolomic effects of Fusarium mycotoxins and MDP supplementation in early-lactation dairy cows.

Author Response

Dear Authors,

Thank you for your detailed and thoughtful responses to my previous comments, and for the revisions made to the manuscript. I appreciate the considerable effort that has gone into addressing the points raised. Below, I offer a few follow-up remarks and minor suggestions for clarification.

Authors: we would like to thank the reviewer for having appreciated the revised manuscript. We would like to also thank the reviewer for improving the overall quality of this manuscript. We have done the minor modifications/revisions required for more clarification.

  1. Analytical methods for mycotoxin quantification

To clarify, my original comment on the analytical methods was not intended to question the validity, accuracy, or accreditation of the HPLC-FLD and GC-MS methods you used. I fully accept that these methods are validated, routinely applied in your laboratory, and suitable for determining ZEN, FUM, and DON. My concern was rather about the conceptual and practical choice of using several separate analytical platforms (HPLC-FLD for ZEN and FUM, GC-MS for DON) instead of a single, unified approach (e.g., LC-MS/MS multi-mycotoxin method), which could, in principle, provide more consistency and efficiency for inter-group comparison. Your response now explains that maintaining continuity with an already validated, accredited workflow and with the companion study (Catellani et al., 2025) was a key reason for this choice. I find this rationale acceptable. However, I would kindly suggest that you make this explicit in the Methods section, perhaps by adding one sentence acknowledging that multi-mycotoxin LC-MS/MS methods are available, but that for this long-term feeding trial you deliberately chose to retain the existing validated workflow for reasons of analytical continuity and comparability.

Authors: thank you very much for the clarification. We have added a new sentence in M&M part to justify the utilization of our existing validated workflow for this long-term feeding trials.

  1. Productive, health, and reproductive parameters

Your response provides a very comprehensive summary of the performance and reproductive findings reported in Catellani et al. (2025), and it is clear that the current metabolomic study is intended as a mechanistic follow-up focusing on subclinical and endocrine effects rather than gross production traits. To further assist readers who may not have immediate access to the companion paper, I would kindly suggest adding either:

a very concise summary in the Results or Discussion (e.g., 1–2 sentences per outcome type, with mean values or directions of change), or

a small supplementary table reporting the key productive and reproductive endpoints from Catellani et al. (2025) for the three groups.

This would allow the biological relevance of the metabolomic alterations to be appreciated without requiring the reader to consult the previous article in detail.

Authors: thank you very much for the suggestion. Indeed, in the discussion section we have deeply provided some directions of change of the main outputs from the companion paper Catellani et al. (2025). Particularly, in terms of reproductive performances we report in paragraph 3.1: " This multi-steroid metabolic fingerprint aligns with the physiological trends previously described in the same animals [14]. Specifically, the earlier study reported that cows fed Fusarium-contaminated diets showed a slower increase in milk progesterone concentrations, fewer corpora lutea (CLs) detected by ultrasound, and a higher proportion of postpartum anestrus cows, especially among primiparous animals. These physiological alterations are compatible with delayed resumption of cyclicity, as confirmed by survival analysis, and a higher tendency for the first dominant follicle (F1) to regress or become cystic rather than ovulate, a pattern that parallels the accumulation of ZEN-derived α-ZEL and β-ZEL metabolites interfering with key steroidogenic enzymes (3α/3β-HSD), which play pivotal roles in progesterone and androgen synthesis [25,26]. By linking these physiological observations to specific metabolic markers, the present plasma metabolomics analysis extends the previous findings by providing molecular-level evidence that dietary Fusarium toxins not only mimic estrogenic effects but also shift the balance of steroid biosynthesis, clearance, and conjugation. The discriminant accumulation of con-jugated estrogen metabolites, together with the decrease of progestins and active androgens, supports the notion that these mycotoxins can alter the delicate balance between estrogenic stimulation and luteal function, particularly during early lactation when cows are under metabolic stress. Taken together, these results indicate that the untargeted metabolomic approach can uncover metabolic patterns potentially related to subtle endocrine alterations, which may be consistent with the physiological delay in cyclicity previously observed. This approach therefore contributes to linking conventional re-productive monitoring with molecular-level metabolic insights." Regarding the access to the companion paper, it was published under Open Access license, therefore it is fully downloadable from several database (e.g., ScienceDirect).

  1. Rationale for using early-lactation dairy cows and single time-point sampling

Your expanded explanation of why early-lactation cows are a relevant and sensitive model, despite ruminal detoxification mechanisms, is convincing, and I agree that the added text in the Introduction/Discussion strengthens the biological context. Likewise, explicitly acknowledging the single time-point design as a limitation and emphasising the need for longitudinal sampling in future studies is appropriate.

 Authors: thank you very much for this comment. As already mentioned in the revised Conclusions section:

" Moreover, the cross-sectional design does not allow assessment of temporal dynamics. Overall, the results suggest that metabolomics can detect subtle, systemic signatures of Fusarium mycotoxin exposure in dairy cows and indicate a potential protective influence of the MDP under field-relevant contamination levels. Future studies incorporating longitudinal sampling, additional physiological and clinical endpoints, and an MDP-only treatment arm will be necessary to confirm the biological significance and mechanistic basis of these preliminary observations."

Overall, I feel that the manuscript has improved and that most of my initial concerns have been satisfactorily addressed. With the minor clarifications suggested above—particularly regarding the explicit rationale for the analytical strategy—the paper will provide a more transparent and robust account of the metabolomic effects of Fusarium mycotoxins and MDP supplementation in early-lactation dairy cows.

Reviewer 3 Report

Comments and Suggestions for Authors

authors have revised sufficiently and paper can be accepted

Author Response

Authors: thank you very much for appreciating the revised version of the manuscript. 

Reviewer 4 Report

Comments and Suggestions for Authors

I mostly agree with the authors' responses. It was not entirely clear from the manuscript that this is a continuation of a previous study, which explains the multiple citations. Information related to the validation of analytical parameters has been added to the Materials and Methods section. I believe that the paper can be published.

Author Response

Authors: we would like to thank the reviewer for having appreciated the revised manuscript.